# Direct then Diffuse:
# Incremental Unsupervised Skill Discovery for State Covering and Goal Reaching

**Pierre-Alexandre Kamienny**[*1,2], **Jean Tarbouriech**[*1,3],
**Sylvain Lamprier**[2], **Alessandro Lazaric**[1], **Ludovic Denoyer**[†1]
[1] Meta AI   [2] Sorbonne University, LIP6/ISIR   [3] Inria Scool

## Abstract

Learning meaningful behaviors in the absence of reward is a difficult problem in reinforcement learning. A desirable and challenging unsupervised objective is to learn a set of diverse skills that provide a thorough coverage of the state space while being directed, i.e., reliably reaching distinct regions of the environment. In this paper, we build on the mutual information framework for skill discovery and introduce UPSIDE, which addresses the coverage-directedness trade-off in the following ways: **1)** We design policies with a decoupled structure of a directed skill, trained to reach a specific region, followed by a diffusing part that induces a local coverage. **2)** We optimize policies by maximizing their number under the constraint that each of them reaches distinct regions of the environment (i.e., they are sufficiently discriminable) and prove that this serves as a lower bound to the original mutual information objective. **3)** Finally, we compose the learned directed skills into a growing tree that adaptively covers the environment. We illustrate in several navigation and control environments how the skills learned by UPSIDE solve sparse-reward downstream tasks better than existing baselines.

## 1 Introduction

Deep reinforcement learning (RL) algorithms have been shown to effectively solve a wide variety of complex problems (e.g., Mnih et al., 2015; Bellemare et al., 2013). However, they are often designed to solve one single task at a time and they need to restart the learning process from scratch for any new problem, even when it is defined on the very same environment (e.g., a robot navigating to different locations in the same apartment). Recently, Unsupervised RL (URL) has been proposed as an approach to address this limitation. In URL, the agent first interacts with the environment without any extrinsic reward signal. Afterward, the agent leverages the experience accumulated during the unsupervised learning phase to efficiently solve a variety of downstream tasks defined on the same environment. This approach is particularly effective in problems such as navigation (see e.g., Bagaria et al., 2021) and robotics (see e.g., Pong et al., 2020) where the agent is often required to readily solve a wide range of tasks while the dynamics of environment remains fixed.

In this paper, we focus on the unsupervised objective of discovering a set of skills that can be used to efficiently solve sparse-reward downstream tasks. In particular, we build on the insight that *mutual information* (MI) between the skills' latent variables and the states reached by them can formalize the dual objective of learning policies that both cover and navigate the environment efficiently. Indeed, maximizing MI has been shown to be a powerful approach for encouraging exploration in RL (Houthooft et al., 2016; Mohamed & Rezende, 2015) and for unsupervised skill discovery (e.g., Gregor et al., 2016; Eysenbach et al., 2019; Achiam et al., 2018; Sharma et al., 2020; Campos et al., 2020). Nonetheless, learning policies that maximize MI is a challenging optimization problem. Several approximations have been proposed to simplify it at the cost of possibly deviating from the original objective of coverage and directedness (see Sect. 4 for a review of related work).

---

[*]equal contribution

[†]Now at Ubisoft La Forge

{pakamienny,jtarbouriech,lazaric}@fb.com, sylvain.lamprier@isir.upmc.fr, ludovic.den@gmail.com

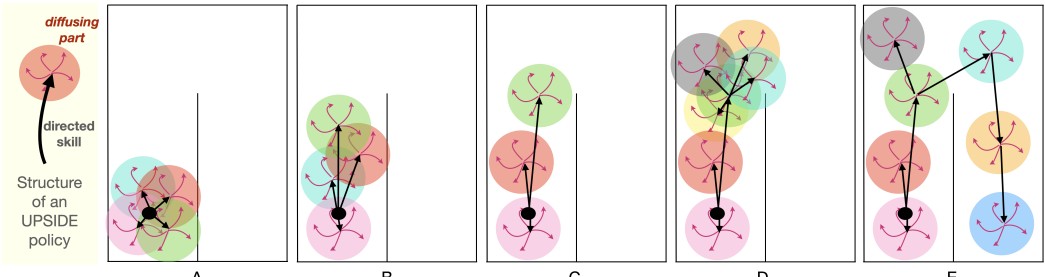

Figure 1: Overview of UPSIDE. The black dot corresponds to the initial state. *(A)* A set of random policies is initialized, each policy being composed of a *directed* part called *skill* (illustrated as a black arrow) and a *diffusing* part (red arrows) which induces a local coverage (colored circles). *(B)* The skills are then updated to maximize the discriminability of the states reached by their corresponding diffusing part (Sect. 3.1). *(C)* The least discriminable policies are iteratively removed while the remaining policies are re-optimized. This is executed until the discriminability of each policy satisfies a given constraint (Sect. 3.2). In this example two policies are consolidated. *(D)* One of these policies is used as basis to add new policies, which are then optimized following the same procedure. For the "red" and "purple" policy, UPSIDE is not able to find sub-policies of sufficient quality and thus they are not expanded any further. *(E)* At the end of the process, UPSIDE has created a tree of policies covering the state space, with skills as edges and diffusing parts as nodes (Sect. 3.3).

In this paper, we propose UPSIDE (*UnsuPervised Skills that dIrect then DiffusE*) to learn a set of policies that can be effectively used to cover the environment and solve goal-reaching downstream tasks. Our solution builds on the following components (Fig. 1):

- *Policy structure (Sect. 3.1, see Fig. 1 (A)).* We consider policies composed of two parts: **1)** a *directed* part, referred to as the *skill*, that is trained to reach a specific region of the environment, and **2)** a *diffusing* part that induces a local coverage around the region attained by the first part. This structure favors coverage and directedness at the level of a single policy.

- *New constrained objective (Sect. 3.2, see Fig. 1 (B) & (C)).* We then introduce a constrained optimization problem designed to maximize the number of policies under the constraint that the states reached by *each* of the diffusing parts are distinct enough (i.e., they satisfy a minimum level of discriminability). We prove that this problem can be cast as a lower bound to the original MI objective, thus preserving its coverage-directedness trade-off. UPSIDE solves it by *adaptively* adding or removing policies to a given initial set, without requiring any prior knowledge on a sensible number of policies.

- *Tree structure (Sect. 3.3, see Fig. 1 (D) & (E)).* Leveraging the directed nature of the skills, UPSIDE effectively composes them to build longer and longer policies organized in a tree structure. This overcomes the need of defining a suitable policy length in advance. Thus in UPSIDE we can consider short policies to make the optimization easier, while composing their skills along a growing tree structure to ensure an adaptive and thorough coverage of the environment.

The combination of these components allows UPSIDE to effectively adapt the number and the length of policies to the specific structure of the environment, while learning policies that ensure coverage and directedness. We study the effectiveness of UPSIDE and the impact of its components in hard-to-explore continuous navigation and control environments, where UPSIDE improves over existing baselines both in terms of exploration and learning performance.

## 2 SETTING

We consider the URL setting where the agent interacts with a Markov decision process (MDP) $M$ with state space $\mathcal{S}$, action space $\mathcal{A}$, dynamics $p(s'|s,a)$, and **no reward**. The agent starts each episode from a designated initial state $s_0 \in \mathcal{S}$.[1] Upon termination of the chosen policy, the agent is then reset to $s_0$. This setting is particularly challenging from an exploration point of view since the agent cannot rely on the initial distribution to cover the state space.

We recall the MI-based unsupervised skill discovery approach (see e.g., Gregor et al., 2016). Denote by $Z$ some (latent) variables on which the policies of length $T$ are conditioned (we assume that $Z$ is categorical for simplicity and because it is the most common case in practice). There are three

---

[1]More generally, $s_0$ could be drawn from any distribution supported over a compact region.

optimization variables: **(i)** the cardinality of $Z$ denoted by $N_Z$, i.e., the number of policies (we write $Z = \{1, \ldots, N_Z\} = [N_Z]$), **(ii)** the parameters $\pi(z)$ of the policy indexed by $z \in Z$, and **(iii)** the policy sampling distribution $\rho$ (i.e., $\rho(z)$ is the probability of sampling policy $z$ at the beginning of the episode). Denote policy $z$'s action distribution in state $s$ by $\pi(\cdot|z, s)$ and the entropy function by $\mathcal{H}$. Let the variable $S_T$ be the random (final) state induced by sampling a policy $z$ from $\rho$ and executing $\pi(z)$ from $s_0$ for an episode. Denote by $p_{\pi(z)}(s_T)$ the distribution over (final) states induced by executing policy $z$, by $p(z|s_T)$ the probability of $z$ being the policy to induce (final) state $s_T$, and let $\overline{p}(s_T) = \sum_{z \in Z} \rho(z) p_{\pi(z)}(s_T)$. Maximizing the MI between $Z$ and $S_T$ can be written as $\max_{N_Z, \rho, \pi} \mathcal{I}(S_T; Z)$, where

$$\mathcal{I}(S_T; Z) = \mathcal{H}(S_T) - \mathcal{H}(S_T|Z) = -\sum_{s_T} \overline{p}(s_T) \log \overline{p}(s_T) + \sum_{z \in Z} \rho(z) \mathbb{E}_{s_T|z} \left[ \log p_{\pi(z)}(s_T) \right]$$

$$= \mathcal{H}(Z) - \mathcal{H}(Z|S_T) = -\sum_{z \in Z} \rho(z) \log \rho(z) + \sum_{z \in Z} \rho(z) \mathbb{E}_{s_T|z} \left[ \log p(z|s_T) \right], \quad (1)$$

where in the expectations $s_T|z \sim p_{\pi(z)}(s_T)$. In the first formulation, the entropy term over states captures the requirement that policies thoroughly cover the state space, while the second term measures the entropy over the states reached by each policy and thus promotes policies that have a directed behavior. Learning the optimal $N_Z$, $\rho$, and $\pi$ to maximize Equation 1 is a challenging problem and several approximations have been proposed (see e.g., Gregor et al., 2016; Eysenbach et al., 2019; Achiam et al., 2018; Campos et al., 2020). Many approaches focus on the so-called *reverse* formulation of the MI (second line of Equation 1). In this case, the conditional distribution $p(z|s_T)$ is usually replaced with a parametric model $q_\phi(z|s_T)$ called the *discriminator* that is trained via a negative log likelihood loss simultaneously with all other variables. Then one can maximize the lower bound (Barber & Agakov, 2004): $\mathcal{I}(S_T; Z) \geq \mathbb{E}_{z \sim \rho(z), \tau \sim \pi(z)} \left[ \log q_\phi(z|s_T) - \log \rho(z) \right]$, where we denote by $\tau \sim \pi(z)$ trajectories sampled from the policy indexed by $z$. As a result, each policy $\pi(z)$ can be trained with RL to maximize the intrinsic reward $r_z(s_T) := \log q_\phi(z|s_T) - \log \rho(z)$.

## 3 THE UPSIDE ALGORITHM

In this section we detail the three main components of UPSIDE, which is summarized in Sect. 3.4.

### 3.1 DECOUPLED POLICY STRUCTURE OF DIRECT-THEN-DIFFUSE

While the trade-off between coverage and directedness is determined by the MI objective, the amount of stochasticity of each policy (e.g., injected via a regularization on the entropy over the actions) has also a major impact on the effectiveness of the overall algorithm (Eysenbach et al., 2019). In fact, while randomness can promote broader coverage, a highly stochastic policy tends to induce a distribution $p_{\pi(z)}(s_T)$ over final states with high entropy, thus increasing $\mathcal{H}(S_T|Z)$ and losing in directedness. In UPSIDE, we define policies with a *decoupled structure* (see Fig. 1 (A)) composed of **a)** a *directed* part (of length $T$) that we refer to as *skill*, with low stochasticity and trained to reach a specific region of the environment and **b)** a *diffusing* part (of length $H$) with high stochasticity to promote local coverage of the states around the region reached by the skill.

Coherently with this structure, the state variable in the conditional entropy in Equation 1 becomes any state reached during the diffusing part (denote by $S_{\text{diff}}$ the random variable) and not just the skill's terminal state. Following Sect. 2 we define an intrinsic reward $r_z(s) = \log q_\phi(z|s) - \log \rho(z)$ and the skill of policy $z$ maximizes the cumulative reward over the states traversed by the diffusing part. Formally, we can conveniently define the objective function:

| | UPSIDE directed skill | UPSIDE diffusing part | VIC policy | DIAYN policy |
|---|---|---|---|---|
| state variable | $S_{\text{diff}}$ | $S_{\text{diff}}$ | $S_T$ | $S$ |
| $\mathcal{J}$ | $\{T, \ldots, T+H\}$ | $\{T, \ldots, T+H\}$ | $\{T\}$ | $\{1, \ldots, T\}$ |
| $(\alpha, \beta)$ | $(1, 0)$ | $(0, 1)$ | $(1, 0)$ | $(1, 1)$ |

Table 1: Instantiation of Equation 2 for each part of an UPSIDE policy, and for VIC (Gregor et al., 2016) and DIAYN (Eysenbach et al., 2019) policies.

$$\max_{\pi(z)} \mathbb{E}_{\tau \sim \pi(z)} \left[ \sum_{t \in \mathcal{J}} \alpha \cdot r_z(s_t) + \beta \cdot \mathcal{H}(\pi(\cdot|z, s_t)) \right], \quad (2)$$

where $\mathcal{J} = \{T, \ldots, T+H\}$ and $\alpha = 1, \beta = 0$ (resp. $\alpha = 0, \beta = 1$) when optimizing for the skill (resp. diffusing part). In words, the skill is incentivized to bring the diffusing part to a discriminable region of the state space, while the diffusing part is optimized by a simple random walk policy (i.e., a stochastic policy with uniform distribution over actions) to promote local coverage around $s_T$.

Table 1 illustrates how UPSIDE's policies compare to other methods. Unlike VIC and similar to DIAYN, the diffusing parts of the policies tend to "push" the skills away so as to reach diverse regions of the environment. The combination of the directedness of the skills and local coverage of the diffusing parts thus ensures that the whole environment can be properly visited with $N_Z \ll |\mathcal{S}|$ policies.[2] Furthermore, the diffusing part can be seen as defining a *cluster of states* that represents the goal region of the directed skill. This is in contrast with DIAYN policies whose stochasticity may be spread over the whole trajectory. This allows us to "ground" the latent variable representations of the policies $Z$ to specific regions of the environment (i.e., the clusters). As a result, maximizing the MI $\mathcal{I}(S_{\text{diff}}; Z)$ can be seen as learning a set of "cluster-conditioned" policies.

## 3.2 A Constrained Optimization Problem

In this section, we focus on how to optimize the number of policies $N_Z$ and the policy sampling distribution $\rho(z)$. The standard practice for Equation 1 is to preset a fixed number of policies $N_Z$ and to fix the distribution $\rho$ to be uniform (see e.g., Eysenbach et al., 2019; Baumli et al., 2021; Strouse et al., 2021). However, using a uniform $\rho$ over a fixed number of policies may be highly suboptimal, in particular when $N_Z$ is not carefully tuned. In App. A.2 we give a simple example and a theoretical argument on how the MI can be ensured to increase by removing skills with low discriminability when $\rho$ is uniform. Motivated by this observation, in UPSIDE we focus on *maximizing the number of policies that are sufficiently discriminable*. We fix the sampling distribution $\rho$ to be uniform over $N$ policies and define the following constrained optimization problem

$$\max_{N \geq 1} N \quad \text{s.t.} \quad g(N) \geq \log \eta, \quad \text{where} \quad g(N) := \max_{\pi, \phi} \min_{z \in [N]} \mathbb{E}_{s_{\text{diff}}} \left[ \log q_\phi(z|s_{\text{diff}}) \right], \quad (\mathcal{P}_\eta)$$

where $q_\phi(z|s_{\text{diff}})$ denotes the probability of $z$ being the policy traversing $s_{\text{diff}}$ during its diffusing part according to the discriminator and $\eta \in (0, 1)$ defines a *minimum discriminability threshold*. By optimizing for $(\mathcal{P}_\eta)$, UPSIDE *automatically adapts* the number of policies to promote coverage, while still guaranteeing that each policy reaches a distinct region of the environment. Alternatively, we can interpret $(\mathcal{P}_\eta)$ under the lens of *clustering*: the aim is to find the largest number of clusters (i.e., the region reached by the directed skill and covered by its associated diffusing part) with a sufficient level of inter-cluster distance (i.e., discriminability) (see Fig. 1). The following lemma (proof in App. A.1) formally links the constrained problem $(\mathcal{P}_\eta)$ back to the original MI objective.

**Lemma 1.** *There exists a value $\eta^\dagger \in (0, 1)$ such that solving $(\mathcal{P}_{\eta^\dagger})$ is equivalent to maximizing a lower bound on the mutual information objective $\max_{N_Z, \rho, \pi, \phi} \mathcal{I}(S_{\text{diff}}; Z)$.*

Since $(\mathcal{P}_{\eta^\dagger})$ is a lower bound to the MI, optimizing it ensures that the algorithm does not deviate too much from the dual covering and directed behavior targeted by MI maximization. Interestingly, Lem. 1 provides a rigorous justification for using a uniform sampling distribution *restricted to the $\eta$-discriminable policies*, which is in striking contrast with most of MI-based literature, where a uniform sampling distribution $\rho$ is defined over the predefined number of policies.

In addition, our alternative objective $(\mathcal{P}_\eta)$ has the benefit of providing a simple *greedy* strategy to optimize the number of policies $N$. Indeed, the following lemma (proof in App. A.1) ensures that starting with $N = 1$ (where $g(1) = 0$) and increasing it until the constraint $g(N) \geq \log \eta$ is violated is guaranteed to terminate with the optimal number of policies.

**Lemma 2.** *The function $g$ is non-increasing in $N$.*

## 3.3 Composing Skills in a Growing Tree Structure

Both the original MI objective and our constrained formulation $(\mathcal{P}_\eta)$ depend on the initial state $s_0$ and on the length of each policy. Although these quantities are usually predefined and only appear implicitly in the equations, they have a crucial impact on the obtained behavior. In fact, resetting after each policy execution unavoidably restricts the coverage to a radius of at most $T + H$ steps around $s_0$. This may suggest to set $T$ and $H$ to large values. However, increasing $T$ makes training the skills more challenging, while increasing $H$ may not be sufficient to improve coverage.

---

[2]Equation 1 is maximized by setting $N_Z = |\mathcal{S}|$ (i.e., $\max_Y \mathcal{I}(X, Y) = \mathcal{I}(X, X) = \mathcal{H}(X)$), where each $z$ represents a goal-conditioned policy reaching a different state, which implies having as many policies as states, thus making the learning particularly challenging.

Instead, we propose to "extend" the length of the policies through composition. We rely on the key insight that *the constraint in $(\mathcal{P}_\eta)$ guarantees that the directed skill of each $\eta$-discriminable policy reliably reaches a specific (and distinct) region of the environment and it is thus re-usable and amenable to composition.* We thus propose to chain the skills so as to reach further and further parts of the state space. Specifically, we build a growing tree, where the root node is a diffusing part around $s_0$, *the edges represent the skills, and the nodes represent the diffusing parts.* When a policy $z$ is selected, the directed skills of its predecessor policies in the tree are executed first (see Fig. 9 in App. B for an illustration). Interestingly, this growing tree structure builds a curriculum on the episode lengths which grow as the sequence $(iT + H)_{i\geq 1}$, thus avoiding the need of prior knowledge on an adequate horizon of the downstream tasks.[3] Here this knowledge is replaced by $T$ and $H$ which are more environment-agnostic and task-agnostic choices as they rather have an impact on the size and shape of the learned tree (e.g., the smaller $T$ and $H$ the bigger the tree).

## 3.4 IMPLEMENTATION

We are now ready to introduce the UPSIDE algorithm, which provides a specific implementation of the components described before (see Fig. 1 for an illustration, Alg. 1 for a short pseudo-code and Alg. 2 in App. B for the detailed version). We first make approximations so that the constraint in $(\mathcal{P}_\eta)$ is easier to estimate. We remove the logarithm from the constraint to have an estimation range of $[0, 1]$ and thus lower variance.[4] We also replace the expectation over $s_{\text{diff}}$ with an empirical estimate $\widehat{q}_\phi^\mathcal{B}(z) = \frac{1}{|\mathcal{B}_z|} \sum_{s\in\mathcal{B}_z} q_\phi(z|s)$, where $\mathcal{B}_z$ denotes a small replay buffer, which we call *state buffer*, that contains states collected during a few rollouts by the diffusing part of $\pi_z$. In our experiments, we take $B = |\mathcal{B}_z| = 10H$. Integrating this in $(\mathcal{P}_\eta)$ leads to

$$\max_{N\geq 1} N \quad \text{s.t.} \quad \max_{\pi,\phi} \min_{z\in[N]} \widehat{q}_\phi^\mathcal{B}(z) \geq \eta, \quad (3)$$

where $\eta$ is an hyper-parameter.[5] From Lem. 2, this optimization problem in $N$ can be solved using the incremental policy addition or removal in Alg. 1 (lines 5 & 9), independently from the number of initial policies $N$.

---

**Algorithm 1:** UPSIDE

**Parameters**: Discriminability threshold $\eta \in (0, 1)$, branching factor $N^{\text{start}}, N^{\text{max}}$.
**Initialize**: Tree $\mathcal{T}$ initialized as a root node 0, policies candidates $\mathcal{Q} = \{0\}$.
**while** $\mathcal{Q} \neq \emptyset$ **do** // `tree expansion`

1  $\quad$ Dequeue a policy $z \in \mathcal{Q}$ and create $N = N^{\text{start}}$ policies $\mathcal{C}(z)$.
2  $\quad$ POLICYLEARNING$(\mathcal{T}, \mathcal{C}(z))$.
3  $\quad$ **if** $\min_{z'\in\mathcal{C}(z)} \widehat{q}_\phi^\mathcal{B}(z') > \eta$ **then**
$\quad$ $\quad$ // `Node addition`
4  $\quad$ $\quad$ **while** $\min_{z'\in\mathcal{C}(z)} \widehat{q}_\phi^\mathcal{B}(z') > \eta$ $\quad$ $\quad$ **and** $N < N^{\text{max}}$ **do**
5  $\quad$ $\quad$ $\quad$ Increment $N = N + 1$ and add one policy to $\mathcal{C}(z)$.
6  $\quad$ $\quad$ $\quad$ POLICYLEARNING$(\mathcal{T}, \mathcal{C}(z))$.
7  $\quad$ **else** // `Node removal`
8  $\quad$ $\quad$ **while** $\min_{z'\in\mathcal{C}(z)} \widehat{q}_\phi^\mathcal{B}(z') < \eta$ $\quad$ $\quad$ **and** $N > 1$ **do**
9  $\quad$ $\quad$ $\quad$ Reduce $N = N - 1$ and remove least discriminable policy from $\mathcal{C}(z)$.
10 $\quad$ $\quad$ $\quad$ POLICYLEARNING$(\mathcal{T}, \mathcal{C}(z))$.
11 $\quad$ Add $\eta$-discriminable policies $\mathcal{C}(z)$ to $\mathcal{Q}$, and to $\mathcal{T}$ as nodes rooted at $z$.

---

We then integrate the optimization of Equation 3 into an adaptive tree expansion strategy that incrementally composes skills (Sect. 3.3). The tree is initialized with a root node corresponding to a policy only composed of the diffusing part around $s_0$. Then UPSIDE iteratively proceeds through the following phases: **(Expansion)** While policies/nodes can be expanded according to different ordering rules (e.g., a FIFO strategy), we rank them in descending order by their discriminability (i.e., $\widehat{q}_\phi^\mathcal{B}(z)$), thus favoring the expansion of policies that reach regions of the state space that are not too saturated. Given a candidate leaf $z$ to expand from the tree, we introduce new policies by adding a set $\mathcal{C}(z)$ of $N = N^{\text{start}}$ nodes rooted at node $z$ (line 2, see also steps *(A)* and *(D)* in Fig. 1). **(Policy learning)** The new policies are optimized in three steps (see App. B for details on the POLICYLEARNING subroutine): **i)** sample states from the diffusing parts of the new policies sampled uniformly from $\mathcal{C}(z)$ (state buffers of consolidated policies in $\mathcal{T}$ are kept in memory), **ii)** update the discriminator and compute the discriminability $\widehat{q}_\phi^\mathcal{B}(z')$ of new policies $z' \in \mathcal{C}(z)$, **iii)**

---

[3]See e.g., the discussion in Mutti et al. (2021) on the "importance of properly choosing the training horizon in accordance with the downstream-task horizon the policy will eventually face."

[4]While Gregor et al. (2016); Eysenbach et al. (2019) employ rewards in the log domain, we find that mapping rewards into $[0, 1]$ works better in practice, as observed in Warde-Farley et al. (2019); Baumli et al. (2021).

[5]Ideally, we would set $\eta = \eta^\dagger$ from Lem. 1, however $\eta^\dagger$ is non-trivial to compute. A strategy may be to solve $(\mathcal{P}_{\eta'})$ for different values of $\eta'$ and select the one that maximizes the MI lower bound $\mathbb{E}\left[\log q_\phi(z|s_{\text{diff}}) - \log \rho(z)\right]$. In our experiments we rather use the same predefined parameter of $\eta = 0.8$ which avoids computational overhead and performs well across all environments.

update the skills to optimize the reward (Sect. 3.1) computed using the discriminator (see step *(B)* in Fig. 1). (**Node adaptation**) Once the policies are trained, UPSIDE proceeds with optimizing $N$ in a greedy fashion. If all the policies in $\mathcal{C}(z)$ have an (estimated) discriminability larger than $\eta$ (lines 3-5), a new policy is tentatively added to $\mathcal{C}(z)$, the policy counter $N$ is incremented, the *policy learning* step is restarted, and the algorithm keeps adding policies until the constraint is not met anymore or a maximum number of policies is attained. Conversely, if every policy in $\mathcal{C}(z)$ does not meet the discriminability constraint (lines 7-9), the one with lowest discriminability is removed from $\mathcal{C}(z)$, the *policy learning* step is restarted, and the algorithm keeps removing policies until all policies satisfy the constraint or no policy is left (see step *(C)* in Fig. 1). The resulting $\mathcal{C}(z)$ is added to the set of *consolidated* policies (line 11) and UPSIDE iteratively proceeds by selecting another node to expand until no node can be expanded (i.e., the *node adaptation* step terminates with $N = 0$ for all nodes) or a maximum number of environment iterations is met.

## 4 RELATED WORK

URL methods can be broadly categorized depending on how the experience of the unsupervised phase is summarized to solve downstream tasks in a zero- or few-shot manner. This includes model-free (e.g., Pong et al., 2020), model-based (e.g., Sekar et al., 2020) and representation learning (e.g., Yarats et al., 2021) methods that build a representative replay buffer to learn accurate estimates or low-dimensional representations. An alternative line of work focuses on discovering a set of skills in an unsupervised way. Our approach falls in this category, on which we now focus this section.

Skill discovery based on MI maximization was first proposed in VIC (Gregor et al., 2016), where the trajectories' final states are considered in the reverse form of Equation 1 and the policy parameters $\pi(z)$ and sampling distribution $\rho$ are simultaneously learned (with a fixed number of skills $N_Z$). DIAYN (Eysenbach et al., 2019) fixes a uniform $\rho$ and weights skills with an action-entropy coefficient (i.e., it additionally minimizes the MI between actions and skills given the state) to push the skills away from each other. DADS (Sharma et al., 2020) learns skills that are not only diverse but also predictable by learned dynamics models, using a generative model over observations and optimizing a forward form of MI $\mathcal{I}(s'; z|s)$ between the next state $s'$ and current skill $z$ (with continuous latent) conditioned on the current state $s$. EDL (Campos et al., 2020) shows that existing skill discovery approaches can provide insufficient coverage and relies on a fixed distribution over states that is either provided by an oracle or learned. SMM (Lee et al., 2019) uses the MI formalism to learn a policy whose state marginal distribution matches a target state distribution (e.g., uniform). Other MI-based skill discovery methods include Florensa et al. (2017); Hansen et al. (2019); Modhe et al. (2020); Baumli et al. (2021); Xie et al. (2021); Liu & Abbeel (2021); Strouse et al. (2021), and extensions in non-episodic settings (Xu et al., 2020; Lu et al., 2020).

While most skill discovery approaches consider a fixed number of policies, a curriculum with increasing $N_Z$ is studied in Achiam et al. (2018); Aubret et al. (2020). We consider a similar discriminability criterion in the constraint in $(\mathcal{P}_\eta)$ and show that it enables to maintain skills that can be readily composed along a tree structure, which can either increase or decrease the support of available skills depending on the region of the state space. Recently, Zhang et al. (2021) propose a hierarchical RL method that discovers abstract skills while jointly learning a higher-level policy to maximize extrinsic reward. Our approach builds on a similar promise of composing skills instead of resetting to $s_0$ after each execution, yet we articulate the composition differently, by exploiting the direct-then-diffuse structure to ground skills to the state space instead of being abstract. Hartikainen et al. (2020) perform unsupervised skill discovery by fitting a distance function; while their approach also includes a directed part and a diffusive part for exploration, it learns only a single directed policy and does not aim to cover the entire state space. Approaches such as DISCERN (Warde-Farley et al., 2019) and Skew-Fit (Pong et al., 2020) learn a goal-conditioned policy in an unsupervised way with an MI objective. As explained by Campos et al. (2020), this can be interpreted as a skill discovery approach with latent $Z = S$, i.e., where each goal state can define a different skill. Conditioning on either goal states or abstract latent skills forms two extremes of the spectrum of unsupervised RL. As argued in Sect. 3.1, we target an intermediate approach of learning "cluster-conditioned" policies. Finally, an alternative approach to skill discovery builds on "spectral" properties of the dynamics of the MDP. This includes eigenoptions (Machado et al., 2017; 2018) and covering options (Jinnai et al., 2019; 2020), and the algorithm of Bagaria et al. (2021) that builds a discrete graph representation which learns and composes spectral skills.

| | Bottleneck Maze | | U-Maze | |
|---|---|---|---|---|
| RANDOM | 29.17 | (±0.57) | 23.33 | (±0.57) |
| DIAYN-10 | 17.67 | (±0.57) | 14.67 | (±0.42) |
| DIAYN-20 | 23.00 | (±1.09) | 16.67 | (±1.10) |
| DIAYN-50 | 30.00 | (±0.72) | 25.33 | (±1.03) |
| DIAYN-curr | 18.00 | (±0.82) | 15.67 | (±0.87) |
| DIAYN-hier | 38.33 | (±0.68) | 49.67 | (±0.57) |
| EDL-10 | 27.00 | (±1.41) | 32.00 | (±1.19) |
| EDL-20 | 31.00 | (±0.47) | 46.00 | (±0.82) |
| EDL-50 | 33.33 | (±0.42) | 52.33 | (±1.23) |
| SMM-10 | 19.00 | (±0.47) | 14.00 | (±0.54) |
| SMM-20 | 23.67 | (±1.29) | 14.00 | (±0.27) |
| SMM-50 | 28.00 | (±0.82) | 25.00 | (±1.52) |
| Flat UPSIDE-10 | 40.67 | (±1.50) | 43.33 | (±2.57) |
| Flat UPSIDE-20 | 47.67 | (±0.31) | 55.67 | (±1.03) |
| Flat UPSIDE-50 | 51.33 | (±1.64) | 57.33 | (±0.31) |
| **UPSIDE** | **85.67** | (±1.93) | **71.33** | (±0.42) |

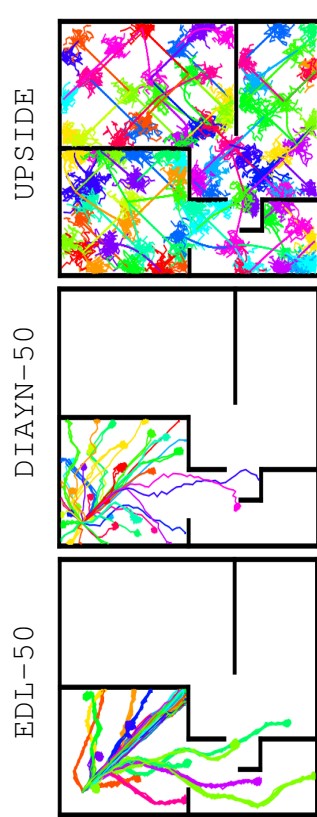

Table 2: Coverage on Bottleneck Maze and U-Maze: UPSIDE covers significantly more regions of the discretized state space than the other methods. The values represent the number of buckets that are reached, where the $50 \times 50$ space is discretized into 10 buckets per axis. To compare the global coverage of methods (and to be fair w.r.t. the amount of injected noise that may vary across methods), we roll-out for each model its associated deterministic policies.

Figure 2: Policies learned on the Bottleneck Maze (see Fig. 14 in App. C for the other methods): contrary to the baselines, UPSIDE successfully escapes the bottleneck region.

## 5 EXPERIMENTS

Our experiments investigate the following questions: **i)** Can UPSIDE incrementally cover an unknown environment while preserving the directedness of its skills? **ii)** Following the unsupervised phase, how can UPSIDE be leveraged to solve sparse-reward goal-reaching downstream tasks? **iii)** What is the impact of the different components of UPSIDE on its performance?

We report results on navigation problems in continuous 2D mazes[6] and on continuous control problems (Brockman et al., 2016; Todorov et al., 2012): Ant, Half-Cheetah and Walker2d. We evaluate performance with the following tasks: **1)** "coverage" which evaluates the extent to which the state space has been covered during the unsupervised phase, and **2)** "unknown goal-reaching" whose objective is to find and reliably reach an unknown goal location through fine-tuning of the policy. We perform our experiments based on the SaLinA framework (Denoyer et al., 2021).

We compare UPSIDE to different baselines. First we consider DIAYN-$N_Z$ (Eysenbach et al., 2019), where $N_Z$ denotes a fixed number of skills. We introduce two new baselines derived from DIAYN: a) DIAYN-curr is a curriculum variant where the number of skills is automatically tuned following the same procedure as in UPSIDE, similar to Achiam et al. (2018), to ensure sufficient discriminability, and b) DIAYN-hier is a hierarchical extension of DIAYN where the skills are composed in a tree as in UPSIDE but without the diffusing part. We also compare to SMM (Lee et al., 2019), which is similar to DIAYN but includes an exploration bonus encouraging the policies to visit rarely encountered states. In addition, we consider EDL (Campos et al., 2020) with the assumption of the available state distribution oracle (since replacing it by SMM does not lead to satisfying results in presence of bottleneck states as shown in Campos et al., 2020). Finally, we consider the RANDOM

---

[6]The agent observes its current position and its actions (in $[-1, +1]$) control its shift in $x$ and $y$ coordinates. We consider two topologies of mazes illustrated in Fig. 2 with size $50 \times 50$ such that exploration is non-trivial. The Bottleneck maze is a harder version of the one in Campos et al. (2020, Fig. 1) whose size is only $10 \times 10$.

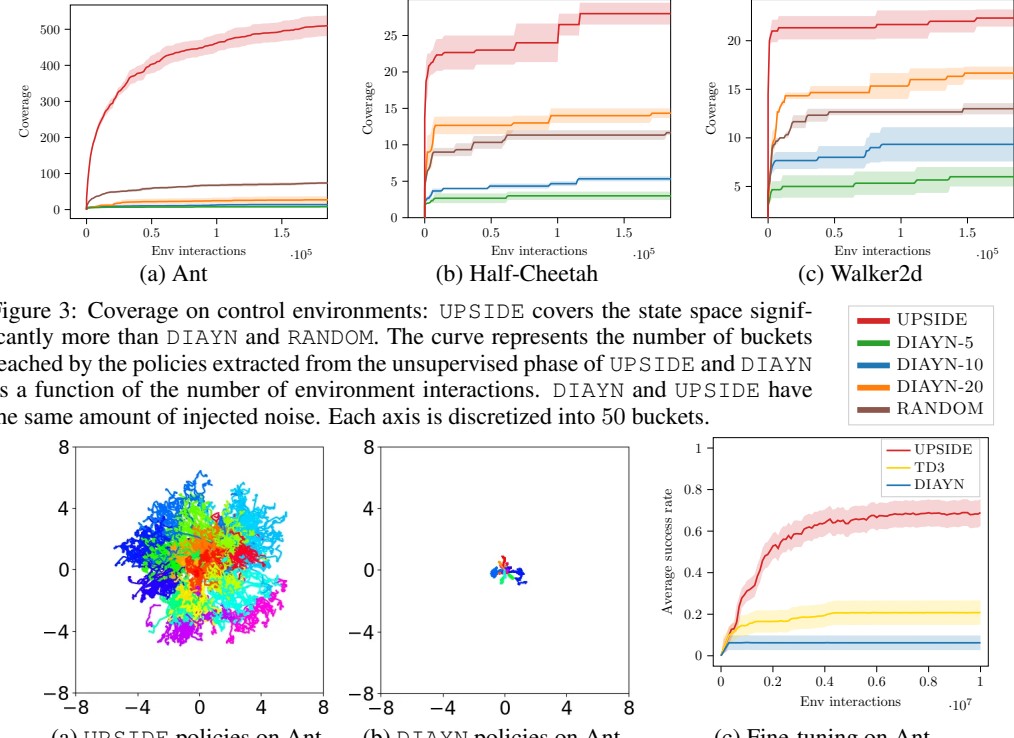

Figure 3: Coverage on control environments: `UPSIDE` covers the state space significantly more than `DIAYN` and `RANDOM`. The curve represents the number of buckets reached by the policies extracted from the unsupervised phase of `UPSIDE` and `DIAYN` as a function of the number of environment interactions. `DIAYN` and `UPSIDE` have the same amount of injected noise. Each axis is discretized into 50 buckets.

Figure 4: (a) & (b) Unsupervised phase on Ant: visualization of the policies learned by `UPSIDE` and `DIAYN-20`. We display only the final skill and the diffusing part of the `UPSIDE` policies. (c) Downstream tasks on Ant: we plot the average success rate over 48 unknown goals (with sparse reward) that are sampled uniformly in the $[-8, 8]^2$ square (using stochastic roll-outs) during the fine-tuning phase. `UPSIDE` achieves higher success rate than `DIAYN-20` and `TD3`.

policy, which samples actions uniformly in the action space. We use `TD3` as the policy optimizer (Fujimoto et al., 2018) though we also tried `SAC` (Haarnoja et al., 2018) which showed equivalent results than `TD3` with harder tuning. Similar to e.g., Eysenbach et al. (2019); Bagaria & Konidaris (2020), we restrict the observation space of the discriminator to the cartesian coordinates $(x, y)$ for Ant and $x$ for Half-Cheetah and Walker2d. All algorithms were ran on $T_{\max} = 1e7$ unsupervised environment interactions in episodes of size $H_{\max} = 200$ (resp. 250) for mazes (resp. for control). For baselines, models are selected according to the cumulated intrinsic reward (as done in e.g., Strouse et al., 2021), while `UPSIDE`, `DIAYN-hier` and `DIAYN-curr` are selected according to the highest number of $\eta$-discriminable policies. On the downstream tasks, we consider `ICM` (Pathak et al., 2017) as an additional baseline. See App. C for the full experimental details.

**Coverage.** We analyze the coverage achieved by the various methods following an unsupervised phase of at most $T_{\max} = 1e7$ environment interactions. For `UPSIDE`, we report coverage for the skill and diffusing part lengths $T = H = 10$ in the continuous mazes (see App. D.4 for an ablation on the values of $T, H$) and $T = H = 50$ in control environments. Fig. 2 shows that `UPSIDE` manages to cover the near-entirety of the state space of the bottleneck maze (including the top-left room) by creating a tree of directed skills, while the other methods struggle to escape from the bottleneck region. This translates quantitatively in the coverage measure of Table 2 where `UPSIDE` achieves the best results. As shown in Fig. 3 and 4, `UPSIDE` clearly outperforms `DIAYN` and `RANDOM` in state-coverage of control environments, for the same number of environment interactions. In the Ant domain, traces from `DIAYN` (Fig. 4b) and discriminator curves in App. D.3 demonstrate that even though `DIAYN` successfully fits 20 policies by learning to take a few steps then hover, it fails to explore the environment. In Half-Cheetah and Walker2d, while `DIAYN` policies learn to fall on the agent's back, `UPSIDE` learns to move forward/backward on its back through skill composition.

**Unknown goal-reaching tasks.** We investigate how the tree of policies learned by `UPSIDE` in the unsupervised phase can be used to tackle goal-reaching downstream tasks. All unsupervised methods follow the same protocol: given an unknown[7] goal $g$, i) we sample rollouts over

---

[7]Notice that if the goal was known, the learned discriminator could be directly used to identify the most promising skill to fine-tune.

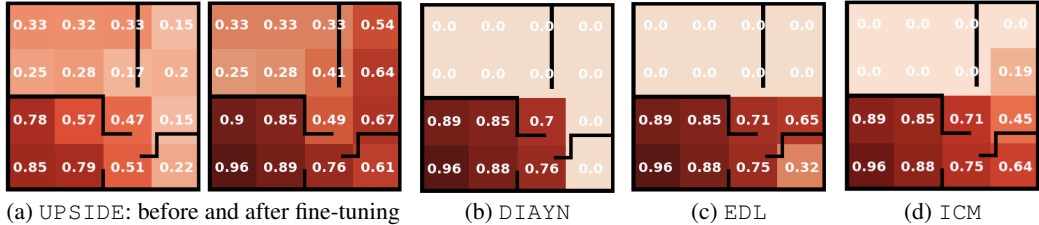

(a) UPSIDE: before and after fine-tuning  (b) DIAYN  (c) EDL  (d) ICM

Figure 5: Downstream task performance on Bottleneck Maze: UPSIDE achieves higher discounted cumulative reward on various unknown goals (See Fig. 15 in App. C for SMM and TD3 performance). From each of the 16 discretized regions, we randomly sample 3 *unknown goals*. For every method and goal seed, we roll-out each policy (learned in the unsupervised phase) during 10 episodes and select the one with largest cumulative reward to fine-tune (with sparse reward $r(s) = \mathbb{I}[\|s - g\|_2 \leq 1]$). Formally, for a given goal $g$ the reported value is $\gamma^\tau \mathbb{I}[\tau \leq H_{\max}]$ with $\tau := \inf\{t \geq 1 : \|s_t - g\|_2 \leq 1\}$, $\gamma = 0.99$ and horizon $H_{\max} = 200$.

the different learned policies, ii) then we select the best policy based on the maximum discounted cumulative reward collected, and iii) we fine-tune this policy (i.e., its sequence of directed skills and its final diffusing part) to maximize the sparse reward $r(s) = \mathbb{I}[\|s - g\|_2 \leq 1]$. Fig. 5 reports the discounted cumulative reward on various goals after the fine-tuning phase. We see that UPSIDE accumulates more reward than the other methods, in particular in regions far from $s_0$, where performing fine-tuning over the entire skill path is especially challenging. In Fig. 6 we see that UPSIDE's fine-tuning can slightly deviate from the original tree structure to improve the goal-reaching behavior of its candidate policy. We also perform fine-tuning on the Ant domain under the same setting. In Fig. 4c, we show that UPSIDE clearly outperforms DIAYN-20 and TD3 when we evaluate the average success rate of reaching 48 goals sampled uniformly in $[-8, 8]^2$. Note that DIAYN particularly fails as its policies learned during the unsupervised phase all stay close to the origin $s_0$.

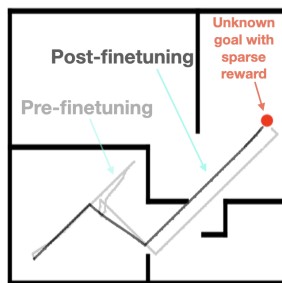

Figure 6: For an unknown goal location, UPSIDE identifies a promising policy in its tree and fine-tunes it.

**Ablative study of the UPSIDE components.** The main components of UPSIDE that differ from existing skill discovery approaches such as DIAYN are: the decoupled policy structure, the constrained optimization problem and the skill chaining via the growing tree. We perform ablations to show that all components are simultaneously required for good performance. First, we compare UPSIDE to flat UPSIDE, i.e., UPSIDE with the tree depth of 1 ($T = 150, H = 50$). Table 2 reveals that the tree structuring is key to improve exploration and escape bottlenecks; it makes the agent learn on smaller and easier problems (i.e., short-horizon MDPs) and mitigates the optimization issues (e.g., non-stationary rewards). However, the diffusing part of flat UPSIDE largely improves the coverage performance over the DIAYN baseline, suggesting that the diffusing part is an interesting structural bias on the entropy regularization that pushes the policies away from each other. This is particularly useful on the Ant environment as shown in Fig. 4. A challenging aspect is to make the skill composition work. As shown in Table 1, DIAYN-hier (a hierarchical version of DIAYN) does not perform as well as UPSIDE by a clear margin. In fact, UPSIDE's direct-then-diffuse decoupling enables both policy re-usability for the chaining (via the directed skills) and local coverage (via the diffusing part). Moreover, as shown by the results of DIAYN-hier on the bottleneck maze, the constrained optimization problem ($\mathcal{P}_\eta$) combined with the diffusing part is crucial to prevent consolidating too many policies, thus allowing a sample efficient growth of the tree structure.

## 6 CONCLUSION AND LIMITATIONS

We introduced UPSIDE, a novel algorithm for unsupervised skill discovery designed to trade off between coverage and directedness and develop a tree of skills that can be used to perform efficient exploration and solve sparse-reward goal-reaching downstream tasks. Limitations of our approach that constitute natural venues for future investigation are: **1)** The diffusing part of each policy could be explicitly trained to maximize local coverage around the skill's terminal state; **2)** UPSIDE assumes a good state representation is provided as input to the discriminator, it would be interesting to pair UPSIDE with effective representation learning techniques to tackle problems with high-dimensional input; **3)** As UPSIDE relies on the ability to reset to establish a root node for its growing tree, it could be relevant to extend the approach in non-episodic environments.

**Acknowledgements** We thank both Evrard Garcelon and Jonas Gehring for helpful discussion.

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

# Appendix

## A  THEORETICAL DETAILS ON SECTION 3

### A.1  PROOFS OF LEMMAS 1 AND 2

**Restatement of Lemma 1.** *There exists a value $\eta^\dagger \in (0,1)$ such that solving $(\mathcal{P}_{\eta^\dagger})$ is equivalent to maximizing a lower bound on the mutual information objective $\max_{N_Z, \rho, \pi, \phi} \mathcal{I}(S_{\mathrm{diff}}; Z)$.*

*Proof.* We assume that the number of available skills is upper bounded, i.e., $1 \leq N_Z \leq N_{\max}$. We begin by lower bounding the negative conditional entropy by using the well known lower bound of Barber & Agakov (2004) on the mutual information

$$-\mathcal{H}(Z|S_{\mathrm{diff}}) = \sum_{z \in Z} \rho(z) \mathbb{E}_{s_{\mathrm{diff}}} \left[ \log p(z|s_{\mathrm{diff}}) \right]$$

$$\geq \sum_{z \in Z} \rho(z) \mathbb{E}_{s_{\mathrm{diff}}} \left[ \log q_\phi(z|s_{\mathrm{diff}}) \right].$$

We now use that any weighted average is lower bounded by its minimum component, which yields

$$-\mathcal{H}(Z|S_{\mathrm{diff}}) \geq \min_{z \in Z} \mathbb{E}_{s_{\mathrm{diff}}} \left[ \log q_\phi(z|s_{\mathrm{diff}}) \right].$$

Thus the following objective is a lower bound on the original objective of maximizing $\mathcal{I}(S_{\mathrm{diff}}; Z)$

$$\max_{N_Z = N, \rho, \pi, \phi} \left\{ \mathcal{H}(Z) + \min_{z \in [N]} \mathbb{E}_{s_{\mathrm{diff}}} \left[ \log q_\phi(z|s_{\mathrm{diff}}) \right] \right\}. \tag{4}$$

Interestingly, the second term in Equation 4 no longer depends on the skill distribution $\rho$, while the first entropy term $\mathcal{H}(Z)$ is maximized by setting $\rho$ to the uniform distribution over $N$ skills (i.e., $\max_\rho \mathcal{H}(Z) = \log(N)$). This enables to simplify the optimization which now only depends on $N$. Thus Equation 4 is equivalent to

$$\max_{N_Z = N} \left\{ \log(N) + \max_{\pi, \phi} \min_{z \in [N]} \mathbb{E}_{s_{\mathrm{diff}}} \left[ \log q_\phi(z|s_{\mathrm{diff}}) \right] \right\}. \tag{5}$$

We define the functions

$$f(N) := \log(N), \qquad g(N) := \max_{\pi, \phi} \min_{z \in [N]} \mathbb{E}_{s_{\mathrm{diff}}} \left[ \log q_\phi(z|s_{\mathrm{diff}}) \right].$$

Let $N^\dagger \in \arg\max_N f(N) + g(N)$ and $\eta^\dagger := \exp g(N^\dagger) \in (0,1)$. We now show that any solution of $(\mathcal{P}_{\eta^\dagger})$ is a solution of Equation 5. Indeed, denote by $N^\star$ the value that optimizes $(\mathcal{P}_{\eta^\dagger})$. First, by validity of the constraint, it holds that $g(N^\star) \geq \log \eta^\dagger = g(N^\dagger)$. Second, since $N^\dagger$ meets the constraint and $N^\star$ is the maximal number of skills that satisfies the constraint of $(\mathcal{P}_{\eta^\dagger})$, by optimality we have that $N^\star \geq N^\dagger$ and therefore $f(N^\star) \geq f(N^\dagger)$ since $f$ is non-decreasing. We thus have

$$\begin{cases} g(N^\star) \geq g(N^\dagger) \\ f(N^\star) \geq f(N^\dagger) \end{cases} \implies f(N^\star) + g(N^\star) \geq f(N^\dagger) + g(N^\dagger).$$

Putting everything together, an optimal solution for $(\mathcal{P}_{\eta^\dagger})$ is an optimal solution for Equation 5, which is equivalent to Equation 4, which is a lower bound of the MI objective, thus concluding the proof. □

**Restatement of Lemma 2.** *The function $g$ is non-increasing in $N$.*

*Proof.* We have that $g(N) := \max_{\pi,q} \min_{z \in [N]} \mathbb{E}_{s \sim \pi(z)}[\log(q(z|s)]$, where throughout the proof we write $s$ instead of $s_{\text{diff}}$ for ease of notation. Here the optimization variables are $\pi \in (\Pi)^N$ (i.e., a set of $N$ policies) and $q : \mathbf{S} \to \Delta(N)$, i.e., a classifier of states to $N$ possible classes, where $\Delta(N)$ denotes the $N$-simplex. For $z \in [N]$, let

$$h_N(\pi, q, z) := \mathbb{E}_{s \sim \pi(z)}[\log(q(z|s)], \qquad f_N(\pi, q) := \min_{z \in [N]} h_N(\pi, q, z).$$

Let $(\pi', q') \in \arg\max_{\pi,q} f_{N+1}(\pi, q)$. We define $\widetilde{\pi} := (\pi'(1), \dots, \pi'(N)) \in (\Pi)^N$ and $\widetilde{q} : \mathbf{S} \to \Delta(N)$ such that $\widetilde{q}(i|s) := q'(i|s)$ for any $i \in [N-1]$ and $\widetilde{q}(N|s) := q'(N|s) + q'(N+1|s)$. Intuitively, we are "discarding" policy $N+1$ and "merging" class $N+1$ with class $N$.

Then it holds that

$$g(N+1) = f_{N+1}(\pi', q') = \min_{z \in [N+1]} h_{N+1}(\pi', q', z) \le \min_{z \in [N]} h_{N+1}(\pi', q', z).$$

Now, by construction of $\widetilde{\pi}, \widetilde{q}$, we have for any $i \in [N-1]$ that $h_{N+1}(\pi', q', i) = h_N(\widetilde{\pi}, \widetilde{q}, i)$. As for class $N$, since $\widetilde{\pi}(N) = \pi'(N)$, by definition of $\widetilde{q}(N|\cdot)$ and from monotonicity of the log function, it holds that $h_{N+1}(\pi', q', N) = \mathbb{E}_{s \sim \pi'(N)}[\log(q'(N|s)]$ satisfies

$$h_{N+1}(\pi', q', N) \le \mathbb{E}_{s \sim \widetilde{\pi}(N)}[\log(\widetilde{q}(N|s)] = h_N(\widetilde{\pi}, \widetilde{q}, N).$$

Hence, we get that

$$\min_{z \in [N]} h_{N+1}(\pi', q', z) \le \min_{z \in [N]} h_N(\widetilde{\pi}, \widetilde{q}, z) = f_N(\widetilde{\pi}, \widetilde{q}) \le g(N).$$

Putting everything together gives $g(N+1) \le g(N)$, which yields the desired result. $\qquad\square$

## A.2 SIMPLE ILLUSTRATION OF THE ISSUE WITH UNIFORM-$\rho$ MI MAXIMIZATION

This section complements Sect. 3.2: we show a simple scenario where **1)** considering both a uniform $\rho$ prior and a fixed skill number $N_Z$ provably leads to suboptimal MI maximization, and where **2)** the UPSIDE strategy of considering a uniform $\rho$ restricted to the $\eta$-discriminable skills can provably increase the MI for small enough $\eta$.

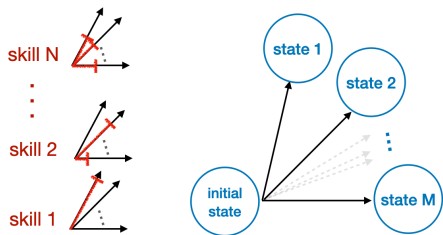

Consider the simple scenario (illustrated on Fig. 7) where the agent has $N$ skills indexed by $n$ and must assign them to $M$ states indexed by $m$. We assume that the execution of each skill deterministically brings it to the assigned state, yet the agent may assign stochastically (i.e., more than one state per skill).

Figure 7: The agent must assign (possibly stochastically) $N$ skills to $M$ states: *under the prior of uniform skill distribution, can the MI with be increased by varying the number of skills $N$?*

(A non-RL way to interpret this is that we want to allocate $N$ balls into $M$ boxes.) Denote by $p_{n,m} \in [0,1]$ the probability that skill $n$ is assigned to state $m$. It must hold that $\forall n \in [N], \sum_m p_{n,m} = 1$. Denote by $\mathcal{I}$ the MI between the skill variable and the assigned state variable, and by $\overline{\mathcal{I}}$ the MI under the prior that the skill sampling distribution $\rho$ is uniform, i.e., $\rho(n) = 1/N$. It holds that

$$\overline{\mathcal{I}}(N, M) = -\sum_n \frac{1}{N} \log \frac{1}{N} + \sum_{n,m} \frac{1}{N} p_{n,m} \log \frac{\frac{1}{N} p_{n,m}}{\sum_n \frac{1}{N} p_{n,m}} = \log N + \frac{1}{N} \sum_{n,m} p_{n,m} \log \frac{p_{n,m}}{\sum_n p_{n,m}}.$$

Let $\overline{\mathcal{I}}^\star(N, M) := \max_{\{p_{n,m}\}} \overline{\mathcal{I}}(N, M)$ and $\{p_{n,m}^\star\} \in \arg\max_{\{p_{n,m}\}} \overline{\mathcal{I}}(N, M)$. We also define the *discriminability* of skill $n$ in state $m$ as

$$q_{n,m} := \frac{p_{n,m}}{\sum_n p_{n,m}},$$

as well as the *minimum discriminability* of the optimal assignment as

$$\eta := \min_n \max_m q_{n,m}^\star.$$

**Lemma 3.** *There exist values of $N$ and $M$ such that the uniform-$\rho$ MI be improved by removing a skill (i.e., by decreasing $N$).*

*Proof.* The following example shows that with $M = 2$ states, it is beneficial for the uniform-$\rho$ MI maximization to go from $N = 3$ to $N = 2$ skills. Indeed, we can numerically compute the optimal solutions and we obtain for $N = 3$ and $M = 2$ that

$$\overline{\mathcal{I}}^{\star}(N = 3, M = 2) \approx 0.918, \qquad p^{\star}_{n,m} = \begin{pmatrix} 0 & 1 \\ 0 & 1 \\ 1 & 0 \end{pmatrix}, \qquad q^{\star}_{n,m} = \begin{pmatrix} 0 & 0.5 \\ 0 & 0.5 \\ 1 & 0 \end{pmatrix}, \qquad \eta = 0.5,$$

whereas for $N = 2$ and $M = 2$,

$$\overline{\mathcal{I}}^{\star}(N = 2, M = 2) = 1, \qquad p^{\star}_{n,m} = \begin{pmatrix} 0 & 1 \\ 1 & 0 \end{pmatrix}, \qquad q^{\star}_{n,m} = \begin{pmatrix} 0 & 1 \\ 1 & 0 \end{pmatrix}, \qquad \eta = 1.$$

As a result, $\overline{\mathcal{I}}^{\star}(N = 2, M = 2) > \overline{\mathcal{I}}^{\star}(N = 3, M = 2)$, which concludes the proof. The intuition why $\overline{\mathcal{I}}^{\star}$ is increased by decreasing $N$ is that for $N = 2$ there is one skill per state whereas for $N = 3$ the skills must necessarily overlap. Note that this contrasts with the original MI (that also optimizes $\rho$) where decreasing $N$ cannot improve the optimal MI. $\qquad \square$

The previous simple example hints to the fact that the value of the minimum discriminability of the optimal assignment $\eta$ may be a good indicator to determine whether to remove a skill. The following more general lemma indeed shows that a sufficient condition for the uniform-$\rho$ MI to be increased by removing a skill is that $\eta$ is small enough.

**Lemma 4.** *Assume without loss of generality that the skill indexed by $N$ has the minimum discriminability $\eta$, i.e., $N \in \arg\min_n \max_m q^{\star}_{n,m}$. Define*

$$\Delta(N, \eta) := \log N - \frac{N-1}{N} \log(N-1) + \frac{1}{N} \log \eta.$$

*If $\Delta(N, \eta) \leq 0$ — which holds for small enough $\eta$ — then removing skill $N$ results in a larger uniform-$\rho$ optimal MI, i.e., $\overline{\mathcal{I}}^{\star}(N, M) < \overline{\mathcal{I}}^{\star}(N-1, M)$.*

*Proof.* It holds that

$$\overline{\mathcal{I}}^{\star}(N, M) = \log N + \frac{1}{N} \left( \sum_{n \in [N-1]} \sum_{m \in [M]} p^{\star}_{n,m} \log q^{\star}_{n,m} + \sum_{m \in [M]} p^{\star}_{n,m} \log \eta \right)$$

$$= \log N - \frac{N-1}{N} \log(N-1)$$

$$+ \frac{N-1}{N} \left( \log(N-1) + \frac{1}{N-1} \sum_{n \in [N-1]} \sum_{m \in [M]} p^{\star}_{n,m} \log q^{\star}_{n,m} \right) + \frac{1}{N} \log \eta$$

$$= \Delta(N, \eta) + \frac{N-1}{N} \overline{\mathcal{I}}^{\star}(N-1, M).$$

As a result, if $\Delta(N, \eta) \leq 0$ then $\overline{\mathcal{I}}^{\star}(N, M) < \overline{\mathcal{I}}^{\star}(N-1, M)$. $\qquad \square$

# B UPSIDE ALGORITHM

## B.1 VISUAL ILLUSTRATIONS OF UPSIDE'S DESIGN MENTIONED IN SECTION 3

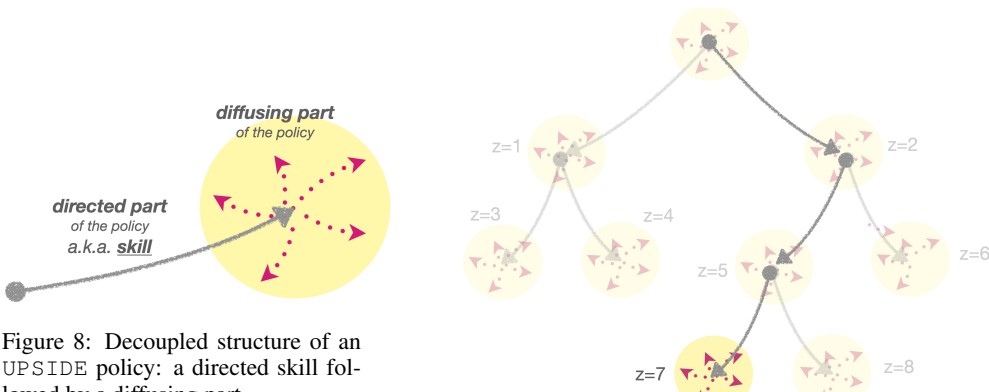

Figure 8: Decoupled structure of an UPSIDE policy: a directed skill followed by a diffusing part.

Figure 9: In the above UPSIDE tree example, executing policy $z = 7$ means sequentially composing the skills of policies $z \in \{2, 5, 7\}$ and then deploying the diffusing part of policy $z = 7$.

## B.2 HIGH-LEVEL APPROACH OF UPSIDE

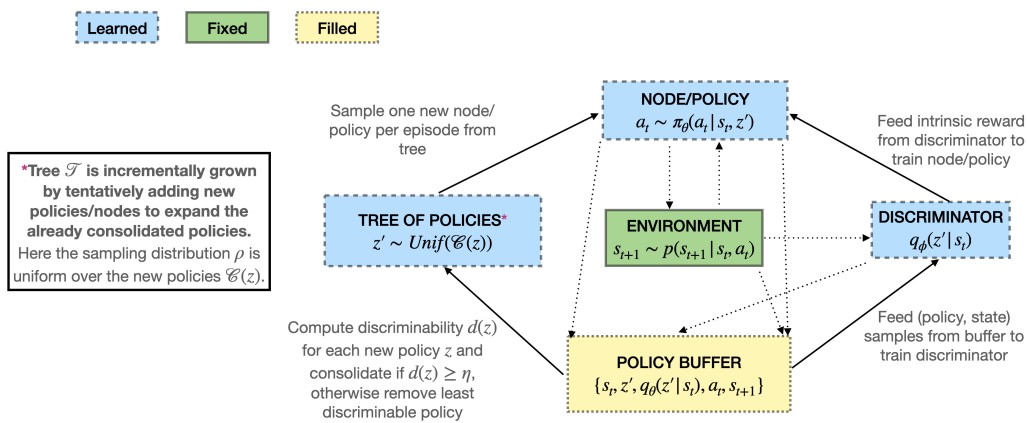

Figure 10: High-level approach of UPSIDE.

## B.3 DETAILS OF ALGORITHM 1

We give in Alg. 2 a more detailed version of Alg. 1 and we list some additional explanations below.

- When optimizing the discriminator, rather than sampling (state, policy) pairs with equal probability for all nodes from the tree $\mathcal{T}$, we put more weight (e.g. $3\times$) on already consolidated policies, which seeks to avoid the new policies from invading the territory of the older policies that were previously correctly learned.
- A replay buffer $\mathcal{B}_{\mathrm{RL}}$ is instantiated at every new expansion (line 2), thus avoiding the need to start collecting data from scratch with the new policies at every POLICYLEARNING call.
- $J$ (line 22) corresponds to the number of policy updates ratio w.r.t. discriminator updates, i.e. for how long the discriminator reward is kept fixed, in order to add stationarity to the reward signal.
- Instead of using a number of iterations $K$ to stop the training loop of the POLICYLEARNING function (line 16), we use a maximum number of environment interactions $K_{\mathrm{steps}}$ for node expansion. Note that this is the same for DIAYN-hier and DIAYN-curr.

---

**Algorithm 2:** Detailed `UPSIDE`

---

**Parameters**: Discriminability threshold $\eta \in (0, 1)$, branching factor $N^{\text{start}}, N^{\text{max}}$
**Initialize**: Tree $\mathcal{T}$ initialized as a root node index by 0, policy candidates $\mathcal{Q} = \{0\}$, state buffers
   $\mathcal{B}_Z = \{0 : [\,]\}$
**while** $\mathcal{Q} \neq \emptyset$ **do** // `tree expansion`

1     Dequeue a policy $z \in \mathcal{Q}$ and create $N = N^{\text{start}}$ nodes $\mathcal{C}(z)$ rooted at $z$ and add new key $z$ to $B_Z$
2     Instantiate new replay buffer $\mathcal{B}_{\text{RL}}$
3     POLICYLEARNING($\mathcal{B}_{\text{RL}}, \mathcal{B}_Z, \mathcal{T}, \mathcal{C}(z)$)
4     **if** $\min_{z' \in \mathcal{C}(z)} d(z') > \eta$ **then** // `Node addition`
5        **while** $\min_{z' \in \mathcal{C}(z)} d(z') > \eta$ **and** $N < N^{max}$ **do**
6           Increment $N = N + 1$ and add one policy to $\mathcal{C}(z)$
7           POLICYLEARNING($\mathcal{B}_{\text{RL}}, \mathcal{B}_Z, \mathcal{T}, \mathcal{C}(z)$)
8     **else** // `Node removal`
9        **while** $\min_{z' \in \mathcal{C}(z)} d(z') < \eta$ **and** $N > 1$ **do**
10       Reduce $N = N - 1$ and remove least discriminable policy from $\mathcal{C}(z)$
11       POLICYLEARNING($\mathcal{B}_{\text{RL}}, \mathcal{B}_Z, \mathcal{T}, \mathcal{C}(z)$)
12     Enqueue in $\mathcal{Q}$ the $\eta$-discriminable nodes $\mathcal{C}(z)$

13 POLICYLEARNING(Replay buffer $\mathcal{B}_{\text{RL}}$, State buffers $\mathcal{B}_Z$, Tree $\mathcal{T}$, policies to update $Z_U$)
14 **Optimization parameters:** patience $K$, policy-to-discriminator update ratio $J$, $K_{\text{discr}}$ discriminator
   update epochs, $K_{\text{pol}}$ policy update epochs
15 **Initialize**: Discriminator $q_\phi$ with $|\mathcal{T}|$ classes
16 **for** $K$ iterations **do** // `Training loop`
17     For all $z' \in Z_U$, clear $\mathcal{B}_Z[z']$ then collect and add $B$ states from the diffusing part of $\pi(z')$ to it
18     Train the discriminator $q_\phi$ for $K_{\text{discr}}$ steps with dataset $\bigcup_{z' \in \mathcal{T}} \mathcal{B}_Z[z']$.
19     Compute discriminability $d(z') = \hat{q}_\phi^{\mathcal{B}}(z') = \frac{1}{|\mathcal{B}_{z'}|} \sum_{s \in \mathcal{B}_{z'}} q_\phi(z'|s)$ for all $z' \in Z_U$
20     **if** $\min_{z' \in Z_U} d(z') > \eta$ **then** // `Early stopping`
21        **Break**
22     **for** $J$ iterations **do**
23        For all $z' \in Z_U$, sample a trajectory from $\pi(z')$ and add to replay buffer $\mathcal{B}_{\text{RL}}$
24        For all $z' \in Z_U$, update policy $\pi'_z$ for $K_{\text{pol}}$ steps on replay buffer $\mathcal{B}_{\text{RL}}$ to optimize the
   discriminator reward as in Sect. 3.1 keeping skills from parent policies fixed
25 Compute discriminability $d(z')$ for all $z' \in Z_U$

---

- The state buffer size $B$ needs to be sufficiently large compared to $H$ so that the state buffers of each policy represent well the distribution of the states generated by the policy's diffusing part. In practice we set $B = 10H$.
- In POLICYLEARNING, we add $K_{\text{initial}}$ random (uniform) transitions to the replay buffer for each newly instantiated policies.
- Moreover, in POLICYLEARNING, instead of sampling uniformly the new policies we sample them in a round robin fashion (i.e., one after the other), which can be simply seen as a variance-reduced version of uniform sampling.

### B.4   ILLUSTRATION OF THE EVOLUTION OF UPSIDE'S TREE ON A WALL-FREE MAZE

See Fig. 11.

### B.5   ILLUSTRATION OF EVOLUTION OF UPSIDE'S TREE ON THE BOTTLENECK MAZE

See Fig. 12.

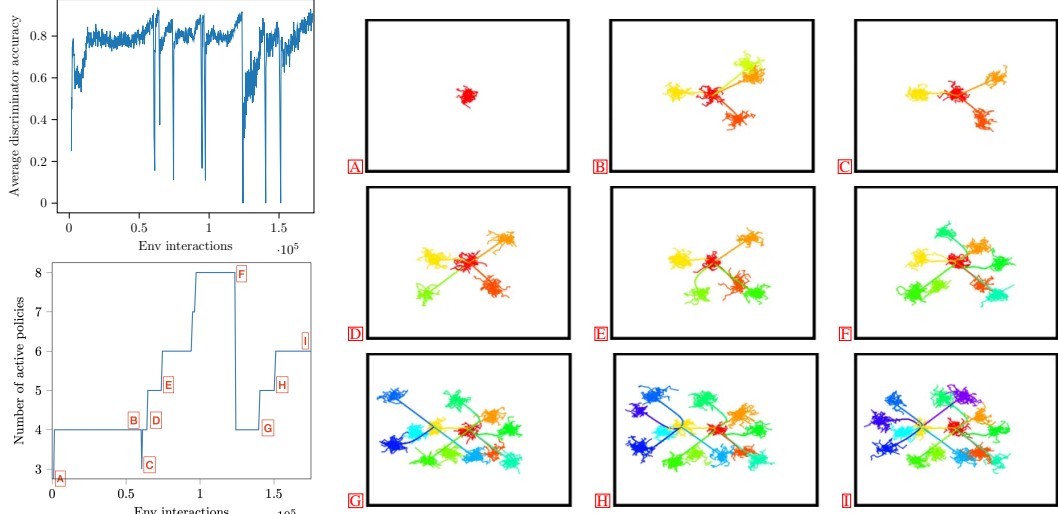

Figure 11: **Fine-grained evolution of the tree structure on a wall-free maze with $N^{\text{start}} = 4$ and $N^{\text{max}} = 8$.** The environment is a wall-free continuous maze with initial state $s_0$ located at the center of the maze. Image A represents the diffusing part around $s_0$. In image B, $N^{\text{start}} = 4$ policies are trained, yet one of them (in lime yellow) is not sufficiently discriminable, thus it is pruned, resulting in image C. A small number of interactions is enough to ensure that the three policies are $\eta$-discriminable (image C). In image D, a fourth policy (in green) is able to become $\eta$-discriminable. New policies are added, trained and $\eta$-discriminated from 5 policies (image E) to $N^{\text{max}} = 8$ policies (image F). Then a policy (in yellow) is expanded with $N^{\text{start}} = 4$ policies (image G). They are all $\eta$-discriminable so additional policies are added (images H, I, . . . ). The process continues until convergence or until time-out (as done here). On the left, we plot the number of active policies (which represents the number of policies that are being trained at the current level of the tree) as well as the average discriminator accuracy over the active policies.

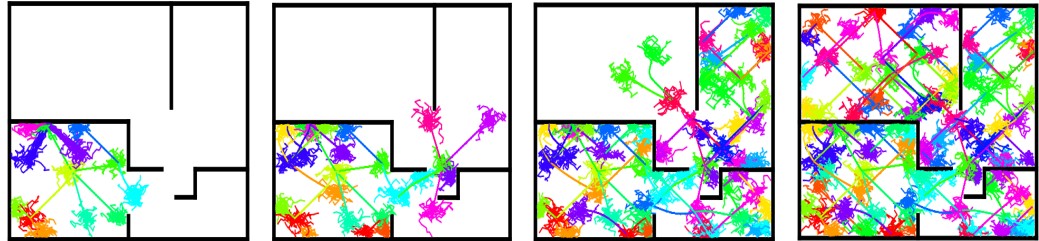

Figure 12: Incremental expansion of the tree learned by UPSIDE towards unexplored regions of the state space in the Bottleneck Maze.

## C EXPERIMENTAL DETAILS

### C.1 BASELINES

**DIAYN-$N_Z$.** This corresponds to the original DIAYN algorithm (Eysenbach et al., 2019) where $N_Z$ is the number of skills to be learned. In order to make the architecture more similar to UPSIDE, we use distinct policies for each skill, i.e. they do not share weights as opposed to Eysenbach et al. (2019). While this may come at the price of sample efficiency, it may also help put lesser constraint on the model (e.g. gradient interference).

**DIAYN-curr.** We augment DIAYN with a curriculum that enables to be less dependent on an adequate tuning of the number of skills of DIAYN. We consider the curriculum of UPSIDE where we start learning with $N^{\text{start}}$ policies during a period of time/number of interactions. If the configuration satisfies the discriminablity threshold $\eta$, a skill is added, otherwise a skill is removed or learning stopped (as in Alg. 2, lines 5-12). Note that the increasing version of this curriculum is similar to the one proposed in VALOR (Achiam et al., 2018, Sect. 3.3). In our experiments, we use $N^{\text{start}} = 1$.

**DIAYN-hier.** We extend `DIAYN` through the use of a hierarchy of directed skills, built following the `UPSIDE` principles. The difference between `DIAYN-hier` and `UPSIDE` is that the discriminator reward is computed over the entire directed skill trajectory, while it is guided by the diffusing part for `UPSIDE`. This introduced baseline can be interpreted as an ablation of `UPSIDE` without the decoupled structure of policies.

**SMM.** We consider `SMM` (Lee et al., 2019) as it is state-of-art in terms of coverage, at least on long-horizon control problems, although Campos et al. (2020) reported its poor performance in hard-to-explore bottleneck mazes. We tested the regular `SMM` version, i.e. learning a state density model with a VAE, yet we failed to make it work on the maze domains that we consider. As we use the cartesian $(x, y)$ positions in maze domains, learning the identity function on two-dimensional input data is too easy with a VAE, thus preventing the benefits of using a density model to drive exploration. In our implementation, the exploration bonus is obtained by maintaining a multinomial distribution over "buckets of states" obtained by discretization (as in our coverage computation), resulting in a computation-efficient implementation that is more stable than the original VAE-based method. Note that the state distribution is computed using states from past-but-recent policies as suggested in the original paper.

**EDL.** We consider `EDL` (Campos et al., 2020) with the strong assumption of an available state distribution oracle (since replacing it by `SMM` does not lead to satisfying results in presence of bottleneck states as shown in Campos et al., 2020, page 7: "We were unable to explore this type of maze effectively with `SMM`"). In our implementation, the oracle samples states uniformly in the mazes avoiding the need to handle a complex exploration, but this setting is not realistic when facing unknown environments.

## C.2 ARCHITECTURE AND HYPERPARAMETERS

The architecture of the different methods remains the same in all our experiments, except that the number of hidden units changes across considered environments. For `UPSIDE`, flat `UPSIDE` (i.e., `UPSIDE` with a tree depth of 1), `DIAYN`, `DIAYN-curr`, `DIAYN-hier` and `SMM` the multiple policies do not share weights, however `EDL` policies all share the same network because of the constraint that the policy embedding $z$ is learnt in a supervised fashion with the VQ-VAE rather than the unsupervised RL objective. We consider decoupled actor and critic optimized with the `TD3` algorithm (Fujimoto et al., 2018) though we also tried `SAC` (Haarnoja et al., 2018) which showed equivalent results than `TD3` with harder tuning.[8] The actor and the critic have the same architecture that processes observations with a two-hidden layers (of size 64 for maze environments and 256 for control environments) neural networks. The discriminator is a two-hidden (of size 64) layer model with output size the number of skills in the tree.

**Common (for all methods and environments) optimization hyper-parameters:**

- Discount factor: $\gamma = 0.99$
- $\sigma_{\text{TD3}} = \{0.1, 0.15, 0.2\}$
- Q-functions soft updates temperature $\tau = 0.005$
- Policy Adam optimizer with learning rate $lr_{\text{pol}} = \{1e^{-3}, 1e^{-4}\}$
- policy inner epochs $K_{\text{pol}} = \{10, 100\}$
- policy batch size $B_{\text{pol}} = \{64\}$
- Discriminator delay: $J = \{1, 10\}$
- Replay buffer maximum size: $1e6$
- $K_{\text{initial}} = 1e3$

We consider the same range of hyper-parameters in the downstream tasks.

---

[8]For completeness, we report here the performance of `DIAYN-SAC` in the continuous mazes: `DIAYN-SAC` with $N_Z = 10$ on Bottleneck maze: 21.0 (± 0.50); on U-maze: 17.5 (± 0.75), to compare with `DIAYN-TD3` with $N_Z = 10$ on Bottleneck maze: 17.67 (± 0.57); on U-maze: 14.67 (± 0.42). We thus see that `DIAYN-SAC` fails to cover the state space, performing similarly to `DIAYN-TD3` (albeit over a larger range of hyperparameter search, possibly explaining the slight improvement).

**`UPSIDE`, `DIAYN` and `SMM` variants (common for all environments) optimization hyper-parameters:**

- Discriminator batch size $B_{\text{discr}} = 64$
- Discriminator Adam optimizer with learning rate $lr_{\text{discr}} = \{1e^{-3}, 1e^{-4}\}$
- discriminator inner epochs $K_{\text{discr}} = \{10, 100\}$
- Discriminator delay: $J = \{1, 10\}$
- State buffer size $B = 10H$ where the diffusing part length $H$ is environment-specific.

**`EDL` optimization hyper-parameters:** We kept the same as Campos et al. (2020). The VQ-VAE's architecture consists of an encoder that takes states as an input and maps them to a code with 2 hidden layers with 128 hidden units and a final linear layer, and the decoder takes the code and maps it back to states also with 2 hidden layers with 128 hidden units. It is trained on the oracle state distribution, then kept fixed during policy learning. Contrary to `UPSIDE`, `DIAYN` and `SMM` variants, the reward is stationary.

- $\beta_{\text{commitment}} = \{0.25, 0.5\}$
- VQ-VAE's code size 16
- VQ-VAE batch size $B_{\text{vq-vae}} = \{64, 256\}$
- total number of epochs: 5000 (trained until convergence)
- VQ-VAE Adam optimizer with learning rate $lr_{\text{vq-vae}} = \{1e^{-3}, 1e^{-4}\}$

**Maze specific hyper-parameters:**

- $K_{\text{steps}} = 5e4$ (and in time 10 minutes)
- $T = H = 10$
- Max episode length $H_{\max} = 200$
- Max number of interactions $T_{\max} = 1e^7$ during unsupervised pre-training and downstream tasks.

**Control specific optimization hyper-parameters:**

- $K_{\text{steps}} = 1e5$ (and in time 1 hour)
- $T = H = 50$
- Max episode length $H_{\max} = 250$
- Max number of interactions $T_{\max} = 1e^7$ during unsupervised pre-training and downstream tasks.

Note that hyperparameters are kept fixed for the downstream tasks too.

### C.3 EXPERIMENTAL PROTOCOL

We now detail the experimental protocol that we followed, which is common for both `UPSIDE` and baselines, on all environments. It consists in the following three stages:

**Unsupervised pre-training phase.** Given an environment, each algorithm is trained without any extrinsic reward on $N_{\text{unsup}} = 3$ seeds which we call *unsupervised seeds* (to account for the randomness in the model weights' initialization and environment stochasticity if present). Each training lasts for a maximum number of $T_{\max}$ environment steps (split in episodes of length $H_{\max}$). This protocol actually favors the baselines since by its design, `UPSIDE` may decide to have fewer environment interactions than $T_{\max}$ thanks to its termination criterion (triggered if it cannot fit any more policies); for instance, all baselines where allowed $T_{\max} = 1e7$ on the maze environments, but `UPSIDE` finished at most in $1e6$ environment steps fitting in average 57 and 51 policies respectively for the Bottleneck Maze and U-Maze.

**Model selection.** For each unsupervised seed, we tune the hyper-parameters of each algorithm according to a certain performance metric. For the baselines, we consider the cumulated intrinsic reward (as done in e.g., Strouse et al., 2021) averaged over stochastic roll-outs. For `UPSIDE`, `DIAYN-hier` and `DIAYN-curr`, the model selection criterion is the number of consolidated policies, i.e., how many policies were $\eta$-discriminated during their training stage. For each method, we thus have as many models as seeds, i.e. $N_{\text{unsup}}$.

**Downstream tasks.** For each algorithm, we evaluate the $N_{\text{unsup}}$ selected models on a set of tasks. All results on downstream tasks will show a performance averaged over the $N_{\text{unsup}}$ seeds.

- **Coverage.** We evaluate to which extent the state space has been covered by discretizing the state space into buckets (10 per axis on the continuous maze domains) and counting how many buckets have been reached. To compare the global coverage of methods (and also to be fair w.r.t. the amount of injected noise that may vary across methods), we roll-out for each model its associated deterministic policies.

- **Fine-tuning on goal-reaching task.** We consider goal-oriented tasks in the discounted episodic setting where the agent needs to reach some unknown goal position within a certain radius (i.e., the goal location is unknown until it is reached once) and with sparse reward signal (i.e., reward of 1 in the goal location, 0 otherwise). The environment terminates when goal is reached or if the number of timesteps is larger than $H_{\max}$. The combination of *unknown goal location and sparse reward* makes the exploration problem very challenging, and calls upon the ability to first cover (for goal finding) and then navigate (for reliable goal reaching) the environment efficiently. To evaluate performance in an exhaustive manner, we discretize the state space into $B_{\text{goal}} = 14$ buckets and we randomly sample $N_{\text{goal}} = 3$ from each of these buckets according to what we call *goal seeds* (thus there are $B_{\text{goal}} \times N_{\text{goal}} = 10$ different goals in total). For every goal seed, we initialize each algorithm with the set of policies learned during the unsupervised pre-training. We then roll-out each policy during $N_{\text{explo}}$ episodes to compute the cumulative reward of the policy, and select the best one to fine-tune. On UPSIDE, we complete the selected policy (of length denoted by $L$) by replacing the diffusing skill with a skill whose length is the remaining number of interactions left, i.e. $H_{\max} - L$. The ability of selecting a good policy is intrinsically linked to the coverage performance of the model, but also to few-shot adaptation. Learning curves and performance are averaged over *unsupervised seeds*, *goal seeds*, and over roll-outs of the stochastic policy. Since we are in the discounted episodic setting, fine-tuning makes sense, to reach as fast as possible the goal. This is particularly important as UPSIDE, because of its tree policy structure, can reach the goal sub-optimally w.r.t the discount. On the maze environments, we consider all unsupervised pre-training baselines as well as "vanilla" baselines trained from scratch during the downstream tasks: TD3 (Fujimoto et al., 2018) and ICM (Pathak et al., 2017). In the Ant environment, we also consider $N_{\text{goal}} = 3$ and $B_{\text{goal}} = 14$ in the $[-8, 8]^2$ square.

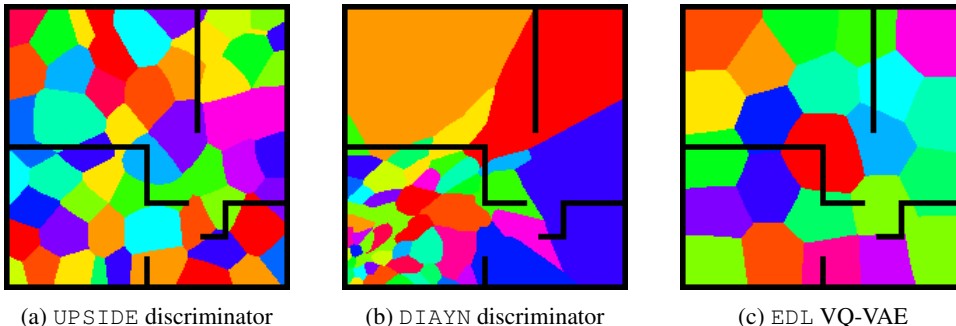

(a) UPSIDE discriminator      (b) DIAYN discriminator      (c) EDL VQ-VAE

Figure 13: Environment divided in colors according to the most likely latent variable $Z$, according to *(from left to right)* the discriminator learned by UPSIDE, the discriminator learned by DIAYN and the VQ-VAE learned by EDL. Contrary to DIAYN, UPSIDE's optimization enables the discriminator training and the policy training to catch up to each other, thus nicely clustering the discriminator predictions across the state space. EDL's VQ-VAE also manages to output good predictions (recall that we consider the EDL version with the strong assumption of the available state distribution oracle, see Campos et al., 2020), yet the skill learning is unable to cover the entire state space due to exploration issues and sparse rewards.

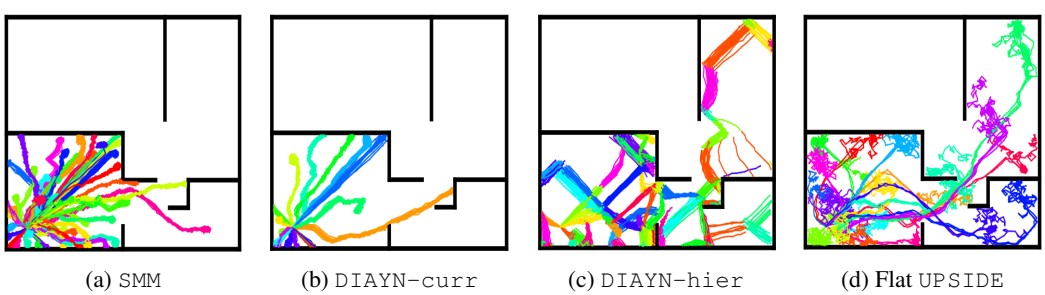

(a) SMM      (b) DIAYN-curr      (c) DIAYN-hier      (d) Flat UPSIDE

Figure 14: Complement to Fig. 2: Visualization of the policies learned on the Bottleneck Maze for the remaining methods.

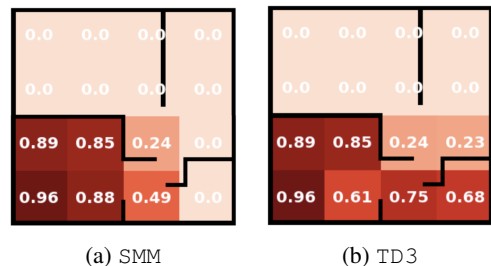

(a) SMM      (b) TD3

Figure 15: Complement of Fig. 5: Heatmaps of downstream task performance after fine-tuning for the remaining methods.

# D   ADDITIONAL EXPERIMENTS

## D.1   ADDITIONAL RESULTS ON BOTTLENECK MAZE

Here we include **1)** Fig. 13 for an analysis of the predictions of the discriminator (see caption for details); **2)** Fig. 14 for the policy visualizations for the remaining methods (i.e., those not reported in Fig. 2; **3)** Fig. 15 for the downstream task performance for the remaining methods (i.e., those not reported in Fig. 5).

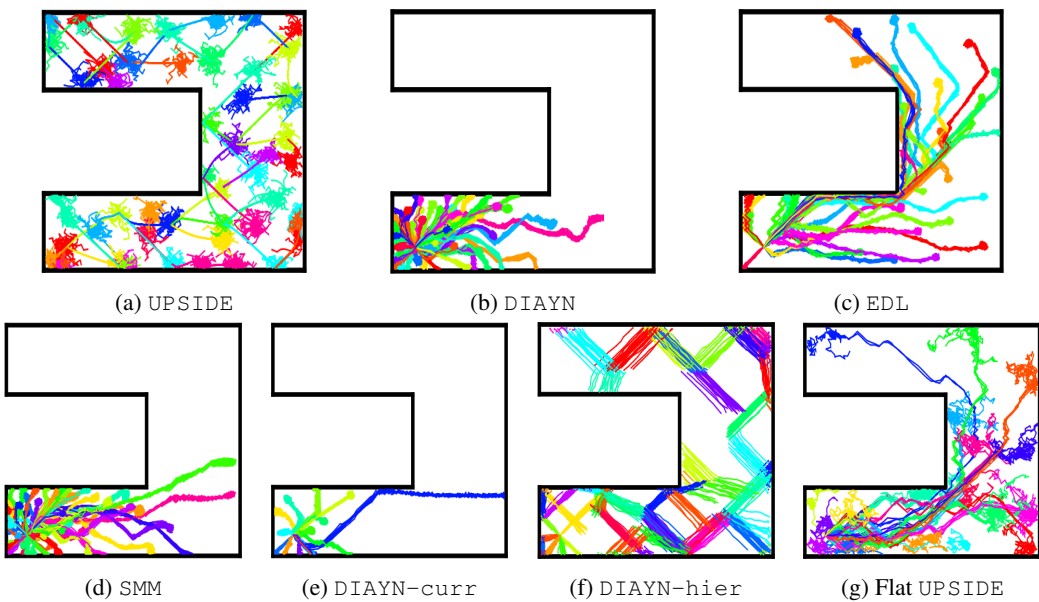

Figure 16: Visualization of the policies learned on U-Maze. This is the equivalent of Fig. 2 for U-Maze.

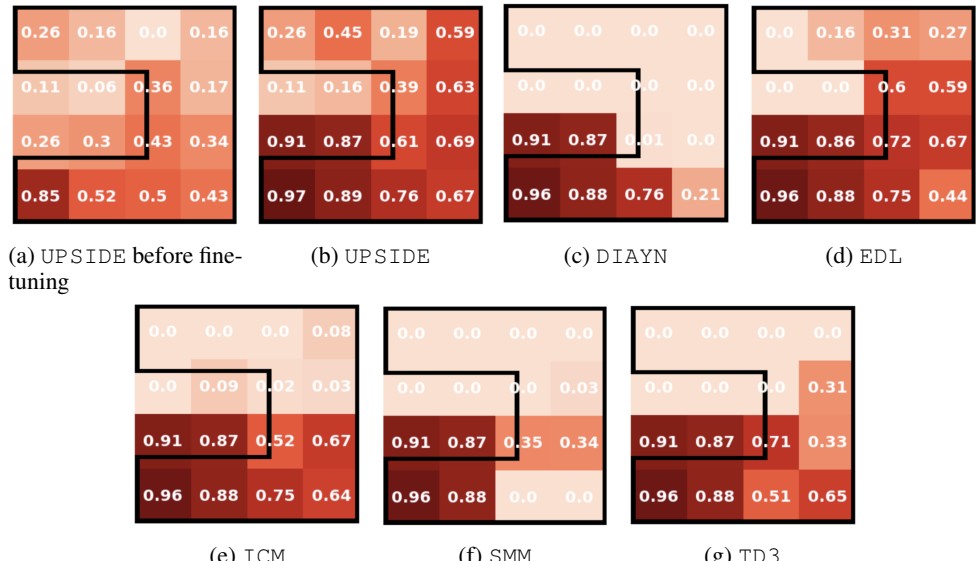

Figure 17: Heat maps of downstream task performance on U-Maze. This is the equivalent of Fig. 5 for U-Maze.

## D.2 ADDITIONAL RESULTS ON U-MAZE

Fig. 16 visualizes the policies learned during the unsupervised phase (i.e., the equivalent of Fig. 2 for the U-Maze), and Fig. 17 reports the heatmaps of downstream task performance (i.e., the equivalent of Fig. 5 for the U-Maze). The conclusion is the same as on the Bottleneck Maze described in Sect. 5: UPSIDE clearly outperforms all the baselines, both in coverage (Fig. 16) and in unknown goal-reaching performance (Fig. 17) in the downstream task phase.

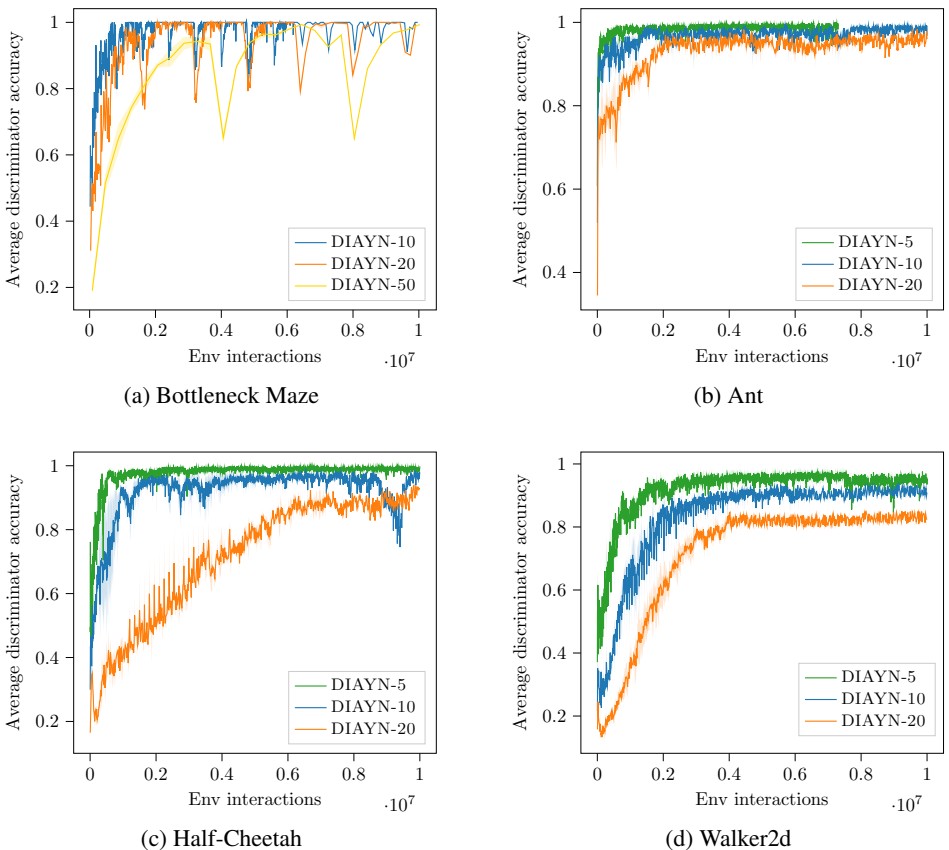

Figure 18: Average discriminability of the DIAYN-$N_Z$ policies. The smaller $N_Z$ is, the easier it is to obtain a close-to-perfect discriminability. However, even for quite large $N_Z$ (50 for mazes and 20 in control environments), DIAYN is able to achieve a good discriminator accuracy, most often because policies learn how to "stop" in some state.

## D.3 ANALYSIS OF THE DISCRIMINABILITY

In Fig. 18 (see caption) we investigate the average discriminability of DIAYN depending on the number of policies $N_Z$.

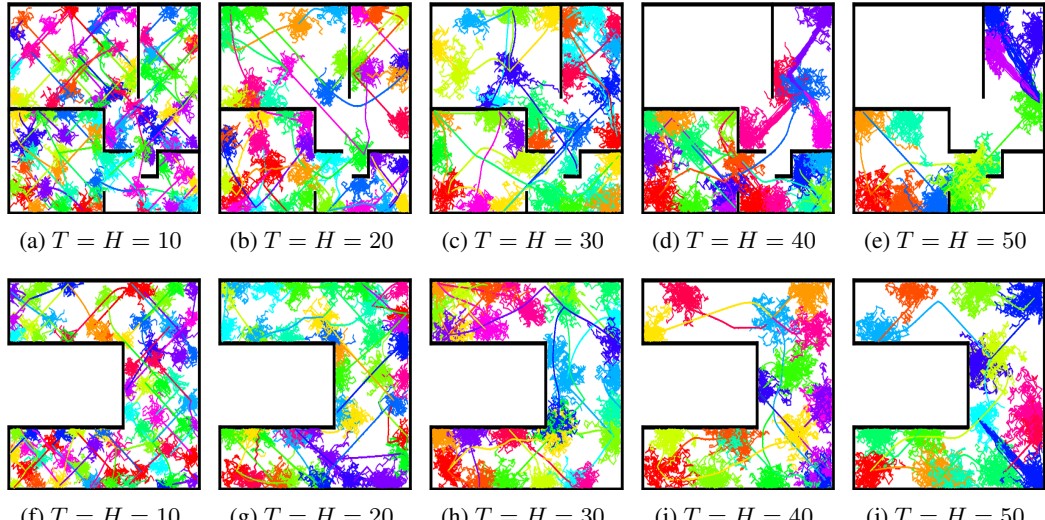

(a) $T = H = 10$    (b) $T = H = 20$    (c) $T = H = 30$    (d) $T = H = 40$    (e) $T = H = 50$

(f) $T = H = 10$    (g) $T = H = 20$    (h) $T = H = 30$    (i) $T = H = 40$    (j) $T = H = 50$

Figure 19: Ablation on the length of UPSIDE policies $(T, H)$: Visualization of the policies learned on the Bottleneck Maze *(top)* and the U-Maze *(bottom)* for different values of $T, H$. *(Right table)* Coverage values (according to the same procedure as in Table 2). Recall that $T$ and $H$ denote respectively the lengths of the directed skill and of the diffusing part of an UPSIDE policy.

| UPSIDE | Bottleneck Maze | U-Maze |
|---|---|---|
| $T = H = 10$ | 85.67   ($\pm$1.93) | 71.33   ($\pm$0.42) |
| $T = H = 20$ | 87.33   ($\pm$0.42) | 67.67   ($\pm$1.50) |
| $T = H = 30$ | 77.33   ($\pm$3.06) | 68.33   ($\pm$0.83) |
| $T = H = 40$ | 59.67   ($\pm$1.81) | 57.33   ($\pm$0.96) |
| $T = H = 50$ | 51.67   ($\pm$0.63) | 58.67   ($\pm$1.26) |

## D.4 ABLATION ON THE LENGTHS $T$ AND $H$ OF THE UPSIDE POLICIES

Our ablation on the mazes in Fig. 19 investigates the sensitiveness of UPSIDE w.r.t. $T$ and $H$, the lengths of the directed skills and diffusing parts of the policies. For the sake of simplicity, we kept $T = H$. It shows that the method is quite robust to reasonable choices of $T$ and $H$, i.e., equal to 10 (as done in all other experiments) but also 20 or 30. Naturally, the performance degrades if $T, H$ are chosen too large w.r.t. the environment size, in particular in the bottleneck maze which requires "narrow" exploration, thus composing disproportionately long skills hinders the coverage. For $T = H = 50$, we recover the performance of flat UPSIDE.

## D.5 FINE-TUNING RESULTS ON HALF-CHEETAH AND WALKER2D

In Sect. 5, we reported the fine-tuning results on Ant. We now focus on Half-Cheetah and Walker2d, and similarly observe that UPSIDE largely outperforms the baselines:

| | UPSIDE | TD3 | DIAYN |
|---|---|---|---|
| Half-Cheetah | 174.93   ($\pm$1.45) | 108.67   ($\pm$25.61) | 0.0   ($\pm$0.0) |
| Walker2d | 46.29   ($\pm$3.09) | 14.33   ($\pm$1.00) | 2.13   ($\pm$0.74) |

We note that the fine-tuning experiment on Half-Cheetah exactly corresponds to the standard Sparse-Half-Cheetah task, by rewarding states where the x-coordinate is larger than 15. Meanwhile, Sparse-Walker2d provides a reward as long as the x-coordinate is larger than 1 and the agent is standing up. Our fine-tuning task on Walker2d is more challenging as we require the x-coordinate to be larger than 4. Yet the agent can be rewarded even if it is not standing up, since our downstream task is purely goal-reaching. We indeed interestingly noticed that UPSIDE may reach the desired goal location yet not be standing up (e.g., by crawling), which may occur as there is no incentive in UPSIDE to remain standing up, only to be discriminable.

