# OpenReview forum: "Direct then Diffuse: Incremental Unsupervised Skill Discovery for State Covering and Goal Reaching"
_ICLR.cc/2022/Conference — ICLR 2022 Poster_

### Official Review · Reviewer_wdBX · 2021-10-19

**Correctness:** 3
**Technical Novelty And Significance:** 2
**Empirical Novelty And Significance:** 3
**Recommendation:** 6
**Confidence:** 4

**Main Review:**


Cons:
- The proposed method is overly complex and contains many ad hoc decisions. For instance, the iteration in Algorithm 1 seems entirely ad hoc to me.
- It is unclear how to adapt the method to a larger or a continuous space of skills, which is important for more realistic applications.
- The theoretical justification for the proposed method is weak. For instance, Lemma 1 shows that the proposed objective is a lower bound on empowerment, but the paper entirely ignores the question of whether the bound is tight (in contrast to all previous work in the area which uses tight bounds). Further, in Sec 3.4, the paper claims that the algorithm is “guaranteed to terminate with the optimal number of policies”, however, the proof ignores the fact that q(z|s) depends on the number of policies. As far as I can tell the algorithm is only guaranteed to find a “locally” optimal solution (i.e. optimal for the corresponding q), not the optimal solution.
- The empirical evaluation unfortunately suffers due to non-standard evaluation metrics. As far as I can tell, the paper does not follow any existing evaluation protocol.

Separately from the cons, I am not convinced that the idea of performing constrained optimization (Eq P_eta) is important for the proposed algorithm. It seems that a very similar algorithm can be obtained by jointly optimizing I(x;z) and N for maximum empowerment.

Pros:
- The paper focuses on the relevant problem of improving coverage in unsupervised skill discovery
- The proposed method significantly outperforms prior work and provides important insights into what properties are important for a successful skill discovery algorithm, such as using discriminability of the final state in the trajectory and stacking skills sequentially for better exploration.
- The paper presents extensive experiments and visualizations which detail the failure points of prior work
- The paper is clearly written and technically relatively strong


**Summary Of The Paper:**

The paper proposes a novel algorithm for learning unsupervised skills based on empowerment. Specifically, direct-and-diffuse policies are proposed that first optimize empowerment and then after a fixed number of episode steps optimize action entropy. This yields policies that go to a certain point in a directed manner and then move around that point at random. Further, the UPSIDE algorithm is based on tracking the policies with high discriminability (i.e. low H(z|x)). The algorithm keeps a set of policies that are currently above a threshold and attempts to add new policies incrementally. Finally, each new policy can be assigned a parent policy, wherein the parent policy is executed to produce the starting state for the new policy. This allows the proposed algorithm to stack policies sequentially and reach goals further away. Empirically, the algorithm achieves good performance on pointmass, cheetah, walker, and ant tasks.

**Summary Of The Review:**

The paper has many issues and in my opinion the proposed algorithm is not viable in more complex domains. However, it is possible that some of the limitations can be lifted, such as using discrete skill variables only. Further, the paper empirically presents good performance on the challenging ant domain and contains several important insights that might be useful for the future researchers. Therefore I am leaning towards accept.

---

> ### Author Response · Authors · 2021-11-19
> **Answer to reviewer wdBX**
>
> Thank you for your valuable feedback which we will incorporate to clarify the paper! Please find our response to your comments below:
>
> **Theoretical justification**: Let us take a step back and retrace our argument. As a starting point, as mentioned by the reviewer, the “ideal” objective would be to jointly optimize I(x;z) over the policies, the distribution over them, and N. However, we identify two issues:
>
> 1) An obvious optimal solution is to set N = |S| (see footnote 3): this suggests that in a large state space, increasing N near-indeterminably is the best strategy.
>
> 2) As observed by Eysenbach et al. (2019, Appendix E), optimizing all variables in I(x;z) is non-trivial (it induces the so-called “skill collapse”), so the common practice is to preset the sampling distribution $\rho$ to be uniform. However, we rigorously prove (Appendix A.2) that under a uniform $\rho$, it may be optimal for the MI to decrease N. This contradicts the simple observation in 1) and illustrates how sensitive the relation between I(x;z) and N is. Moreover, it shows that presetting N (as done in most prior works) may lead to suboptimal MI maximization.
>
> Our alternative objective (P_eta) has the following benefits:
>
> 1) It can optimize N using a simple greedy procedure (see Section 3.4) (i.e., starting with N=1 and increasing until the constraint is violated), where the optimal N remains reasonable thanks to the diffusing part.
> 2) It provides a rigorous justification for using a $\rho$ that is uniform but restricted to the η-discriminable policies (see proof of Lemma 1).
>
> (P_eta) is a lower bound to the original MI objective, which shows that we do not deviate too much from the dual coverage-directedness trade-off. The reviewer argues that “the paper entirely ignores the question of whether the bound is tight (in contrast to all previous work in the area which uses tight bounds)”. We respectfully disagree with the claim that our theoretical justification is looser than prior work. For instance, in DIAYN and most subsequent works, setting a uniform $\rho$ in the MI objective (that should in theory also optimize for $\rho$) also induces a decrease in MI (see e.g., simple example of Appendix A.2) that is completely unaccounted for and may be far from tight.
>
> All in all, we believe that (P_eta) provides an alternative objective that is simple, interpretable (e.g., under the lens of clustering), anchored in the original MI intuition, geared towards N-optimization, and crucially amenable to skill composition (Section 3.3, which is a core feature of UPSIDE).
>
> **Regarding the convergence to the ‘optimal’ N**: The optimization problem indeed depends on multiple variables (number of policies N, policies $\pi$, discriminator $\phi$). For analysis clarity, we focus on studying the evolution of N while $\pi$ and $\phi$ are fixed. Note that in Algorithm 1, $\pi$ and $\phi$ are also optimized simultaneously for each value of N: this is the PolicyLearning step. In practice, there is a natural trade-off in the duration of each PolicyLearning step between sample efficiency (short duration) and quality of the final N value (the longer the duration, the closer the ‘locally’ optimal solution gets to a ‘global’ solution). We will clarify this point in the revised version.
>
> **Evaluation protocol**: Our coverage metric (where we discretize the state space and count the number of visited regions) can be related to the “test-time exploration” metric of SMM, Lee et al. (2019), which computes the percentage of (unknown, randomly generated) goals found, the only difference being that our generation of goals ensures that the entire state space is considered. The downstream task metric on Ant (Fig. 4c) is similar to the success rate considered by prior work (e.g., Lee et al., 2019; Bagaria et al., 2021). The one in the mazes (Fig. 5) is almost the same, the only difference being that we add a discount factor to the sparse reward, which is able to capture the speed at which the agent reaches the desired goal (i.e., the faster it reaches it, the better).
>
> **More complex domains**: We believe that the three algorithmic components of UPSIDE (listed in the abstract) provide relevant insights to further advance our understanding of the current limitations and important features for unsupervised skill discovery. We believe that each UPSIDE component could be amenable to adaptation to more complex domains, which is an exciting and important direction for future investigation. For example, in environments where random exploration does not provide some local coverage, the diffusing part could be explicitly trained to maximize state entropy around the skill’s terminal state, by e.g., using SMM (Lee et al., 2019).

---

> > ### Comment · Reviewer_wdBX · 2021-11-30
> > **wdBX Response**
> >
> > The rebuttal unfortunately does not address my concerns.
> >
> > >  We respectfully disagree with the claim that our theoretical justification is looser than prior work. For instance, in DIAYN and most subsequent works, setting a uniform  in the MI objective (that should in theory also optimize for ) also induces a decrease in MI (see e.g., simple example of Appendix A.2) that is completely unaccounted for and may be far from tight.
> >
> > This is not correct. Gregor'16 learns the skill prior. Regardless, the bound is actually tight for any setting of the prior (but the true MI will be different for different priors). The argument in the appendix has no relation to whether the lower bound is tight: it is instead an argument about which prior distribution yields the highest MI.
> >
> > > In practice, there is a natural trade-off in the duration of each PolicyLearning step between sample efficiency (short duration) and quality of the final N value (the longer the duration, the closer the ‘locally’ optimal solution gets to a ‘global’ solution). We will clarify this point in the revised version.
> >
> > I must be missing the part where this is proven. I also can't see the clarifications in the revised version. If the analysis fixes \pi and \phi, such analysis cannot prove the statement above (since \pi and \phi are not actually fixed). If the analysis does not fix these (this is unclear from the text), it is missing a few additional steps to prove that g(N) is non-increasing.

---

> > > ### Author Response · Authors · 2021-12-01
> > > **Response to reviewer wdBX**
> > >
> > > We thank the reviewer for the discussion and hope that our reply below (which we will incorporate in the final version) clarifies the two remaining interrogations.
> > >
> > > ***
> > > **Point 1.** We agree with the reviewer's comments and do not believe that it contradicts our motivation, which we can summarize as follows:
> > > - The ideal objective would be $F_1 := \max_{N,\rho,\pi,\phi} \textrm{MI}$.
> > > - In VIC, Gregor et al. (2016) fix the number of skills N and optimize $F_2 := \max_{\rho,\pi,\phi} \textrm{MI}$.
> > > - In DIAYN (Eysenbach et al., 2019) and most subsequent works, the skill prior $\rho$ is also fixed (to uniform, for stability) and the optimization is $F_3 := \max_{\pi,\phi} \textrm{MI}$.
> > >
> > > While $F_1 \geq F_2 \geq F_3$, there is no study on how tight or loose these inequalities are. In fact, Appendix A.2 gives a simple example where fixing both $N$ and $\rho$ strictly decreases the optimal MI.
> > >
> > > Our objective $(P_{\eta})$ builds on an alternative way of lower bounding $F_1$. As mentioned by the reviewer, the gap can similarly be loose w.r.t. $F_1$ (more generally, designing a tractable optimization problem for a deep URL algorithm arguably requires some level of approximation, as directly optimizing $F_1$ is very difficult). Our alternative MI-inspired formulation $(P_{\eta})$ comes with new benefits: it provides a simple and interpretable way to **1)** adapt over the skill number $N$ and **2)** chain skills dynamically. Our experiments show that our approach succeeds in doing both, while also (as a byproduct) empirically improving the coverage-directedness trade-off over prior MI-based URL approaches.
> > >
> > > ***
> > > **Point 2**. To answer the reviewer's second point, we prove below the non-increasingness of the $g$ function which captures the constraint in $(P_{\eta})$ (see also Eq.3). We have that (notice the flip between max and min, apologies for the typo in the paper) $g(N) := \max_{\pi,q} \min_{z \in [N]} \textbf{E}_{s \sim \pi(z)} [ \log(q(z\vert s)]$ (also here we write $s$ instead of $s$_diff for ease of notation).
> > >
> > >  The optimization variables are $\pi \in (\Pi)^N$ (i.e., a set of $N$ policies) and $q: \textbf{S} \rightarrow \Delta(N)$, i.e., a classifier of states to $N$ possible classes ($\Delta(N)$ denotes the $N$-simplex).
> > >
> > > For $z \in [N]$, let:
> > >
> > > $$h_N(\pi,q,z) := \textbf{E}_{s \sim \pi(z)} [ \log(q(z\vert s)],$$
> > >
> > > $$f_N(\pi,q) := \min_{z \in [N]} h_N(\pi,q,z).$$
> > >
> > > Let $(\pi', q') \in \arg\max_{\pi,q} f_{N+1}(\pi,q)$. We define $\widetilde{\pi} := ( \pi'(1), \ldots, \pi'(N) ) \in (\Pi)^N$ and $\widetilde{q}: \textbf{S} \rightarrow \Delta(N)$ such that $\widetilde{q}(i \vert s) := q'(i \vert s)$ for any $i \in [N-1]$ and $\widetilde{q}(N \vert s) := q'(N \vert s) +  q'(N+1 \vert s)$. Intuitively, we are "discarding" policy $N+1$ and "merging" class $N+1$ with class $N$.
> > >
> > > Then it holds that: $$g(N+1) = f_{N+1}(\pi',q') = \min_{z \in [N+1]} h_{N+1}(\pi',q',z) \leq \min_{z \in [N]} h_{N+1}(\pi',q',z).$$ Now, by construction of $\widetilde{\pi},\widetilde{q}$, we have for any $i \in [N-1]$ that
> > > $h_{N+1}(\pi',q',i) = h_N(\widetilde{\pi},\widetilde{q},i)$. As for class $N$, by $\widetilde{\pi}(N) = \pi'(N)$, by definition of $\widetilde{q}(N \vert \cdot)$ and by monotonicity of the log function, it holds that $h_{N+1}(\pi',q',N) = \textbf{E}_{s \sim \pi'(N)} [ \log(q'(N\vert s)]$ satisfies:
> > >
> > > $$h_{N+1}(\pi',q',N) \leq \textbf{E}_{s \sim \tilde{\pi}(N)} [ \log(\widetilde{q}(N\vert s)] = h_N(\widetilde{\pi},\widetilde{q},N).$$
> > >
> > > Hence, we get that: $$\min_{z \in [N]} h_{N+1}(\pi',q',z) \leq \min_{z \in [N]} h_{N}(\widetilde{\pi},\widetilde{q},z) = f_N(\widetilde{\pi},\widetilde{q}) \leq g(N).$$ Putting everything together gives $g(N+1) \leq g(N)$, which yields the desired result.

---

> > > > ### Comment · Reviewer_wdBX · 2021-12-02
> > > > **wdBX Response**
> > > >
> > > > Thank you for providing the proof. The proof appears correct and would improve the paper if included. However, it is still unclear whether Alg 1 converges to the optimal solution. This is because the structure of the skill tree must also be optimized, but the proof ignores that.
> > > >
> > > > In fact, Alg 1 doesn't appear to terminate at all as far as I can understand.
> > > >
> > > > Line 9 of Alg 1 further appears to be contradictory: it might be impossible to remove the least discriminable policy if other policies depend on it. This policy can still be kept in the memory, but that would produce a potentially unbounded memory growth which is undesirable.
> > > >
> > > > Overall, I still believe the paper is not much better than borderline so I will maintain my score.
> > > >
> > > > **MI optimization.** The authors response argues that the proposed method achieves a better value of mutual information by optimizing over the number of skills, which seems possibly correct to me. However, my original concern was instead about how tight is the lower bound in Eq 4 of the appendix, which is simply a lower bound on mutual information of the current policy and skill distribution (rather than the optimal value which is what the author response is about).
> > > >
> > > > Also note the mutual information does not depend on the encoder, which is only a variational approximation.

---

> > > > > ### Author Response · Authors · 2021-12-03
> > > > > **Response to reviewer wdBX**
> > > > >
> > > > > Indeed, as articulated in Section 3.4, the discussed proof holds in the "flat case". This gives us a principled way to optimize the tree structure *locally* level by level, whereas the tree is expanded incrementally, which enables a tractable search in an otherwise combinatorial space.
> > > > >
> > > > > We point out that this strategy does have a natural termination condition: Algorithm 1 stops when there are no more children nodes that can be expanded (see "while $\mathcal{Q} \neq \emptyset$"), which occurs when the state space is saturated and does not allow for additional discriminable-enough policies.
> > > > >
> > > > > In Algorithm 1, whenever the least discriminable policy is removed (line 9), moving from $N$ to $N-1$ policies, we proceed with retraining the remaining policies (rooted at the same node) to ensure that they are optimized for the new configuration (line 10). Each policy has independent weights (with its own parameters), which enables to remove policies without keeping them in memory. This allows us to practically deploy Algorithm 1, and we observe that it yields the desired outcome: for instance, the learned tree manages to incrementally and adaptively cover the Bottleneck Maze which prior work struggled to cover, see Figure 12.
> > > > >
> > > > > Thank you for the thorough discussion whose clarifications points will be incorporated in the paper!

---

### Official Review · Reviewer_H3Jm · 2021-10-28

**Correctness:** 4
**Technical Novelty And Significance:** 3
**Empirical Novelty And Significance:** 3
**Recommendation:** 8
**Confidence:** 4

**Main Review:**

The submission has all components that make up a good paper: it reads well, describes and motivates the problem, positions itself with respect to the prior literature, covers the theory and implementation with necessary details, and compares the prior methods. I don’t have any critical concerns, and only mostly minor comments and suggestions.

I cannot fully parse the objective in Equation 2. Especially the sentence “In words, the skill is incentivized to bring the diffusing part to a discriminable region of the state space.” seems to be incorrect. The reward does incentivize the skills to stay in a discriminable region, but only the discriminability of the end of the episode should matter (not how we got there). In fact, I would expect this objective to lead to skills that try to avoid unvisited states (as those regions are indiscriminable) and the skills should be biased in already visited direction rather than being symmetrically distributed, as is the case for example in Figure 4(a). Can you elaborate on why the chosen reward works?

Regarding the experiments and comparison to DIYAN, I find it strange that the coverage of RANDOM is practically as good as DYIANs, and in the case of Ant, RANDOM actually surpasses DYIAN. The text clarifies that in the Half-Cheetah and Walker2d environments, “DIAYN policies learn to fall on the agent’s back”. This does not seem to be the case in the original DIAYN paper (https://sites.google.com/view/diayn), where the agent especially in the HalfCheetah environment is able to acquire proper gaits. Can you comment on this discrepancy?

What are the axes in Figure 4(a) and (b)? I suppose they are the xy-coordinates, in which case UPSIDE covers the space really well. Just to confirm, the discriminator gets to see the full state, not just these two coordinates?

The sparse reward tasks do not bring much new to the paper because the result is obvious from the coverage plots showing UPSIDE explores well. I would suggest using one of more standard sparse exploration tasks (e.g., SparseHalfCheetah) to better show the benefits and make the result comparable to existing literature.

There is a missing related paper [1], which also learns unsupervised skills by fitting a distance function. They learn only a single directed policy and do not try to cover the entire state space, but their policy also includes a directed part and a diffusive part for exploration.

Equation 1 has a missing maximization operator on the right hand side of the equation.

[1] Hartikainen et al., Dynamical Distance Learning for Semi-Supervised and Unsupervised Skill Discovery, ICLR 2020


**Summary Of The Paper:**

The paper proposes an unsupervised exploration method for reinforcement learning, called UPSIDE, that combines learning of directed skills that enable covering distant states, and a diffusive part that explores locally and helps expand the explored region further. The main contributions are both the topology of the policy (division to tree-structured skills and diffusive parts), a theory for training such policies, and a practical implementation that simplifies some of the technicalities induced by the theory. The experiments illustrate well the assumed benefits and include both toy tasks (a point mass in a maze) as well as more complex tasks such as HalfCheetah and Ant from OpenAI Gym. The experiments also compare to prior methods such as DIYAN and provide ablations of the importance of the different components.


**Summary Of The Review:**

Good paper that has all required components. The proposed algorithm is novel, the paper provides both theoretical analysis as well as a practical version of the algorithm, and the experiments indicate substantial improvement over prior works.

---

> ### Author Response · Authors · 2021-11-19
> **Answer to reviewer H3Jm**
>
> Thank you for appreciating our work! We will incorporate your valuable comments and suggestions (e.g., we will add the reference of Hartikainen et al., 2020). Please find our response to the comments below.
>
> **Objective in Equation 2**: The directed skill optimizes for $\mathbb{E}[\sum_{t=T}^{T+H} r_z(s_t)]$: it is thus rewarded by how much the diffusing part attains a discriminable region of the state space. This has the effect of pushing the skills away from each other. What happens during the directed skill’s trajectory (indexed by $[0, …, T]$) is indeed not taken into account.
>
> The reviewer makes a great point that a key aspect of such intrinsic rewards (i.e., VIC, DIAYN) is that ``two functions – the intrinsic reward of the discriminator and the policy – have to match up’’ (Gregor et al., 2016). Indeed, policies visiting novel states will initially be given an intrinsic reward of q = 1/N due to the discriminator's initial inaccurate (random) predictions. This may not sufficiently encourage skills to explore new regions and lead them to a suboptimal behavior. This challenge has been well documented (see e.g., Gregor et al., 2016; Campos et al., 2020). Here we could argue that UPSIDE partly mitigate this issue (shared by VIC, DIAYN, ...) in the two following ways: 1) First, the skill composition thanks to the tree structure implies that each optimization is performed with a short horizon (of $T+H$), which eases the learning dynamics. 2) Second, the decoupled policy structure entails that the discriminator is trained not only on the terminal state $s_T$ of a novel trajectory (as in VIC), but also on all of the diffusing states $\{s}_{T \leq t \leq T+H}$, which implies a factor $H$ amplification of the training signal for the discriminator at novel regions.
>
> **DIAYN performance**: Our findings of poor performance of DIAYN on control environments seem consistent with e.g., IBOL (Kim et al., 2021). One reason for the discrepancy with the original results is that we perform model selection according to discriminability, instead of e.g., the maximum distance, which we have observed to provide better results on such a specific task. We believe it is important to have an unsupervised model selection criterion instead of a task-specific one.
>
>
> **Axes in Figure 4**: The axes are indeed the (x,y) coordinates. As we mention in Section 5, “Similar to e.g., Eysenbach et al. (2019); Bagaria & Konidaris (2020), we restrict the observation space of the discriminator to the cartesian coordinates (x,y) for Ant and x for Half-Cheetah and Walker2d.”
>
> **Finetuning**: We study the before/after-finetuning performance in the maze environments to isolate the benefits of performing finetuning following the unsupervised phase. We add finetuning results on HalfCheetah and Walker2d:   We similarly observe that UPSIDE largely outperforms the baselines, as we obtain the following results for UPSIDE, TD3 and DIAYN respectively:
>
> Half-Cheetah:  174.93(1.45), 108.67(25.61), 0.0(0.0)
> Walker2d: 46.29 (3.09), 14.33(1.00), 2.13(0.74)
>
>
> As suggested by the reviewer, our additional finetuning experiment on HalfCheetah exactly corresponds to the standard SparseHalfCheetah task (i.e., rewarding states where the x-coordinate is larger than 15). Meanwhile, SparseWalker2d provides a reward as long as the x-coordinate is larger than 1 and the agent is standing up. Our additional finetuning task on Walker2d is more challenging as we require the x-coordinate to be larger than 4. Yet the agent can be rewarded even if it is not standing up, since our downstream task is purely goal-reaching. We indeed interestingly noticed that UPSIDE may reach the desired goal location yet not be standing up (e.g., by crawling), which may occur as there is no incentive in UPSIDE to remain standing up, only to be discriminable.
>
> Reference: Kim et al. (2021): Unsupervised Skill Discovery with Bottleneck Option Learning

---

> > ### Comment · Reviewer_H3Jm · 2021-11-28
> > **Response**
> >
> > Thank you for the detailed explanation that answers my remaining questions. I have no further comments. Great work!

---

### Official Review · Reviewer_zwRd · 2021-11-02

**Correctness:** 3
**Technical Novelty And Significance:** 3
**Empirical Novelty And Significance:** 3
**Recommendation:** 6
**Confidence:** 4

**Main Review:**

**Strengths**:
This work takes a novel approach to the skill discovery problem. Instead of directly finding maximal entropy policies that act as skills, this work splits skills into directed and diffusing (exploratory) components. The paper also includes a technique for limiting the number of skills by ensuring they satisfy some minimal discriminability threshold. Finally, the method also includes a method for generating compositions of directed skills via a tree structure and then diffusing from the leaf to explore. Overall the method is novel and is appears to be supported by experimental results.

**Weaknesses**:
I think there are several weaknesses in the presentation and experimentation that would need to be addressed.
1. For the continuous control environments, why are there no finetuning results on the downstream tasks besides Ant? For some reason the authors report coverage and discriminator accuracy for three gym tasks: Ant, Half-Cheetah and Walker but only downstream RL results on Ant. Please report downstream results on the other two tasks as well.
2. Why is DADS not included as a baseline? DADs has been shown to outperform DIAYN on the continuous control tasks and should certainly be compared against.
3. I am slightly confused by the discussion regarding a fixed number of skills N - DIAYN at least doesn’t have this - it just has a skill conditioned policy with z sampled from the uniform distribution.

-   Using DIAYN-Nz doesn’t seem particularly valid to me - you should compare against the original method as well for proper comparison

-   The authors then introduce modified variants of DIAYN (DIAYN-curr) to deal with the fact that it has a limited set of policies - which was not how the algorithm was designed to begin with. Additionally they then switch to TD3 claiming it doesn’t work well. Instead as mentioned above, please comparison against the original DIAYN codebase.

Minor Nitpicks:
- Add some sort of visualizations of the learned skills for the continuous control environments.
- “We optimize policies by maximizing their number under the constraint that each of them reaches distinct regions of the environment” - this sentence needs a wording fix - its a bit unclear in the abstract itself what maximizing their number means.
- Figure 5: what does UPSIDE before/after mean?
- Figure 10 or something like it should go in the main paper (it is very helpful in understanding the main idea of the work)

Overall, I recommend rejection but I am open to updating my score if my concerns are addressed.

**Summary Of The Paper:**

This work proposes a new framework for unsupervised skill discovery termed UPSIDE which aims to learn a fix set of state space covering skills that have a directed and diffusing (noisy) component, are constrained to be sufficiently discriminable and are chained together in a tree structure to enable composition. The paper shows results on grid world and continuous control environments demonstrating the effectiveness of their method.

**Summary Of The Review:**

Overall, the method proposed in this work is novel but the experiments, while showing performance improvements for the proposed method, do have several flaws which I point out in my review. If these issues can be addressed, I would be open to updating my score from rejection.

=====================================UPDATE=========================================================
After discussion with the authors, I have raised my score as my experimental concerns have been addressed.

---

> ### Author Response · Authors · 2021-11-19
> **Answer to reviewer zwRd**
>
> Thank you for your valuable feedback which we will incorporate! Please find our response to your comments below:
>
> **MuJoCo finetuning**: As suggested, we perform fine-tuning experiments on Half-Cheetah and Walker2d in the same way that in the Ant experiment on a far-away goal ($x>=15$ for Half-Cheetah and $x>=4$ for Walker2d). We similarly observe that UPSIDE largely outperforms the baselines, as we obtain the following results for UPSIDE, TD3 and DIAYN respectively:
> Half-Cheetah:  174.93(1.45), 108.67(25.61), 0.0(0.0)
> Walker2d: 46.29 (3.09), 14.33(1.00), 2.13(0.74)
>
> **Baselines on MuJoCo**: We have implemented DADS (Sharma et al., 2020) and in the supplementary material (ant_coverage_with_dads.png), we report a preliminary empirical result of the coverage in the Ant environment (more results will be provided in the future version). We observe that DADS explores the state space better than DIAYN yet it is significantly outperformed by UPSIDE. This shows the relevance of the UPSIDE-specific components of the decoupled skill structure and the skill composition via the tree structure, which have the effect of further pushing the skills away from each other and favoring incremental coverage of the state space. Note that we consider a discrete-latent DADS to align with the other methods that we consider, as also done in Campos et al. (2020). It could be an interesting future direction to extend UPSIDE to continuous latent space and try to be adaptive with respect to the dimension $D$ (in continuous DADS skills are sampled as $z \sim \text{Uniform}(−1, 1)^D$).
>
> **DIAYN details**: We would like to clarify that DIAYN requires a preset number of skills. As mentioned by the authors (https://github.com/ben-eysenbach/sac/blob/master/DIAYN.md): “In our experiments, we fixed the number of skills apriori, and sampled uniformly from this distribution during training”. Thus, DIAYN-Nz does correspond to the original algorithm of Eysenbach et al. (2019), where we report different values for Nz as it can be a quite sensitive hyperparameter.
>
> In addition, we switched our implementation of DIAYN from SAC to TD3 since it showed similar results while being easier to tune. This is coherent with the recent paper proposing a set of unsupervised RL benchmarks by Laskin et al. (2021) who argue that SAC “is prone to suffering from policy entropy collapse” and rather focus on DDPG as backbone RL algorithm (we use TD3 which is an improved version of DDPG). For completeness, we report below the performance of DIAYN-SAC in the continuous mazes:
> DIAYN-SAC with Nz=10 on Bottleneck maze: 21.0 (± 0.50); on U-maze: 17.5 (± 0.75) to compare with DIAYN-TD3 with Nz=10  on Bottleneck maze: 17.67(± 0.57); on U-maze: 14.67 (± 0.42).
> We observe that it fails to cover the state space, performing similarly to DIAYN-TD3 (albeit over a larger range of hyperparameter search, possibly explaining the slight improvement).
>
> **Figure 5: what does UPSIDE before/after mean?** UPSIDE (before) is the “zero-shot” performance of our method following the unsupervised phase, i.e., the performance of the best available skill. UPSIDE (after) is the performance of our method following the fine-tuning phase on the downstream task. This isolates the benefits of the fine-tuning phase on UPSIDE’s performance.
>
> Finally we thank the reviewer for the suggestion to put Figure 10 in the main paper (which we will incorporate in the revised version) and to add visualizations (we are planning to create a website with videos of the learned behaviors).
>
> Reference: Laskin et al. (2021): URLB: Unsupervised Reinforcement Learning Benchmark

---

> > ### Comment · Reviewer_zwRd · 2021-11-22
> > **Response to Authors**
> >
> > I thank the authors for their response and for addressing each of my concerns.
> > I would encourage the authors to include DADS results for all of the experiments run in the main paper and include the new Half-Cheetah and Walker2D results in the main paper as well (with DADS comparison).
> > I will raise my score due to this discussion.

---

### Official Review · Reviewer_d1m1 · 2021-11-02

**Correctness:** 4
**Technical Novelty And Significance:** 3
**Empirical Novelty And Significance:** 3
**Recommendation:** 8
**Confidence:** 3

**Main Review:**

**Strengths**

* This is an interesting approach to improve unsupervised skill discovery in multiple ways: state space coverage, avoiding prespecifying the number of skills to learn, and dynamically extending episode horizons.

  Most skill discovery methods that use the mutual information decomposition of a form of $\mathcal{I}(S; Z) = \mathcal{H}(Z) - \mathcal{H}(Z|S)$ fix the distribution of $z$ and do not encourage exploration. This work suggests a workaround to explore local regions with a decoupled policy structure (direct and diffuse), which I find a meaningful contribution.

  The approach of maximizing the number of skills via the constrained objective and learning new skills on top of existing ones in the tree-structured manner is also an intriguing direction for skill discovery with less predetermined components.

* In comparison with the baseline skill discovery methods, this method shows improved state space coverage and performance empirically (see my comment about the issue with MuJoCo experiments below, though).

* The visualization and analyses provide meaningful insights. They show that the proposed method is effective for learning a set of skills covering the state space on the toy (Maze) environments (Table 2 and Fig.2). They also present how well their method can reach different goals on Bottleneck Maze before and after fine-tuning (Fig.5).

**Weaknesses**

* The diffusing with the simple random walk policy (performing uniform random actions) is one of the most basic strategy for exploration and wouldn't be very effective on many environments, as it does not necessarily lead to good (local) state space coverage. Fig.4 (a) is an example of such case. This is also mentioned by the authors in Sec.6.

* Only a few baseline methods are used for the comparison on the MuJoCo environments (Ant, Half-Cheetah, and Walker2d), considering that those environments are much more complex and closer to real-world scenarios than the Maze environments.

  More importantly, DIAYN (Eysenbach et al., 2019) and RANDOM are not a good choice of baselines for comparing state space coverages in MuJoCo control environments. The authors can check out/consider other works, such as DADS (Sharma et al., 2020) or IBOL (Kim et al., 2021).

**Summary Of The Paper:**

To increase the state space coverage with unsupervised skill discovery, the authors propose UPSIDE. They tackle reaching distant states (direct) and covering neighbor states (diffuse) at different stages. Also, they discover skills by forming a tree, which enables chaining of smaller skills to form far-reaching skills. Also, instead of fixing the number of skills to learn, they maximize the number of skills with a constraint on the output of the discriminator $q_\phi(z|s_{diff})$ and show that it is a lower bound of the mutual information objective. Empirical results on Maze and MuJoCo environments and further analyses are presented.

**Summary Of The Review:**

The authors propose an interesting approach to unsupervised skill discovery tackling multiple issues with some existing prior methods. They also provide meaningful empirical results and analyses.

---

> ### Author Response · Authors · 2021-11-19
> **Answer to d1m1**
>
> Thank you for the valuable comments and for appreciating our work! Please find below our response to the comments which we will incorporate in the paper:
>
> **Diffusing part with RW**: The random walk is the easiest strategy to implement the diffusing part. Given that in UPSIDE we can leverage the hierarchical structure to “extend” the horizon of the skills, we can afford using a relatively small length H for the diffusing part, which guarantees a sufficient  local coverage in the environments that we consider. Otherwise, the diffusing part could be explicitly trained to maximize state entropy around the skill’s terminal state, by e.g., using SMM (Lee et al., 2019) or other MaxEnt-based algorithms. We leave the study of this variant for future work.
>
>
> **Baselines on MuJoCo**: We have implemented DADS (Sharma et al., 2020) and in the supplementary material (ant_coverage_with_dads.png), we report a preliminary empirical result of the coverage in the Ant environment (more results will be provided in the future version). We observe that DADS explores the state space better than DIAYN yet it is significantly outperformed by UPSIDE. This shows the relevance of the UPSIDE-specific components of the decoupled skill structure and the skill composition via the tree structure, which have the effect of further pushing the skills away from each other and favoring incremental coverage of the state space. Note that we consider a discrete-latent DADS to align with the other methods that we consider, as also done in Campos et al. (2020). It could be an interesting future direction to extend UPSIDE to continuous latent space and try to be adaptive with respect to the dimension $D$ (in continuous DADS skills are sampled as $z \sim \text{Uniform}(−1, 1)^D$).

---

### Decision · Program_Chairs · 2022-01-20

**Decision:**

Accept (Poster)

**Comment:**

The authors propose UPSIDE, a method for improving state coverage in unsupervised skill learning. "Direct-and-diffuse" policies concatenate phases of directed behaviour followed by dithering to improve neighbourhood coverage in the region of state space visited, trained with a discriminability objective to ensure diversity in region coverage, and composing these skills into a tree structure to further improve coverage.

Reviewers generally found the submission highly novel and well written, were encouraged by the experimental results, and were unanimous on the importance of the problem addressed. Reviewers d1m1 and wdBX praised the visualizations as adding considerable insight. d1m1 raised the concern that a random walk in the second stage would not work in some environments (which the authors pointed out can be overcome by the tree method) and suggested DADS was a better baseline than DIAYN (also mentioned by zwRd, with which the authors agreed, and proceeded to implement). zwRd expressed concerns about gaps in the finetuning results (addressed by the authors in follow-up) as well as details of the DIAYN setup (also addressed to zwRd's satisfaction). H3Jm's minor concerns around the text and equations, figures and DIAYN experimental results were fully addressed in discussion. wdBX was the reviewer with the most substantive concerns around the apparent complexity, ad hoc design, scalability and lack of theoretical grounding. After discussion (which resulted in the authors drafting a proof in the comments, an ICLR first for this AC) wdBX believed that the paper would be improved by additions proposed, but remained "borderline" in their evaluation.

The work presents a compelling, if somewhat complicated (and therefore perhaps unsatisfying, to some), method improving state coverage in unsupervised skill learning. Research in contemporary machine learning often takes the form of one piece of work pushing the boundaries significantly with a complex algorithm with further work dissecting and refining the ideas therein, reducing them to their essence. Viewed through this lens, the paper in question presents several ideas that all form parts of the overall method which appear likely to serve to inspire and motivate future work. With all reviewers leaning accept either strongly or weakly, there is little doubt in the AC's mind that this paper should be accepted, and widely read by researchers interested in unsupervised skill acquisition.